# Watermarking Graph Neural Networks via Explanations for Ownership Protection

## Abstract

Graph Neural Networks (GNNs) are the mainstream method to learn pervasive graph data and are widely deployed in industry, making their intellectual property valuable. However, protecting GNNs from unauthorized use remains a challenge. Watermarking, which embeds ownership information into a model, is a potential solution. However, existing watermarking methods have two key limitations: First, almost all of them focus on non-graph data, with watermarking GNNs for complex graph data largely unexplored. Second, the *de facto* backdoor-based watermarking methods pollute training data and induce ownership ambiguity through intentional misclassification. Our explanation-based watermarking inherits the strengths of backdoor-based methods (e.g., robust to watermark removal attacks), but avoids data pollution and eliminates intentional misclassification. In particular, our method learns to embed the watermark in GNN explanations such that this unique watermark is statistically distinct from other potential solutions, and ownership claims must show statistical significance to be verified. We theoretically prove that, even with full knowledge of our method, locating the watermark is an NP-hard problem. Empirically, our method manifests robustness to removal attacks like fine-tuning and pruning. By addressing these challenges, our approach marks a significant advancement in protecting GNN intellectual property.

## 1 Introduction

Graph Neural Networks (GNNs) (Scarselli et al., 2008; Kipf & Welling, 2017; Hamilton et al., 2018; Veličković et al., 2018) are widely used for tasks involving pervasive graph-structured data, such as social network analysis, bioinformatics, and recommendation systems (Zhang et al., 2021; Zhou et al., 2020). Various giant companies have integrated GNNs into their systems or open-sourced their GNN frameworks: Amazon uses GNNs to analyze user behavior patterns for product recommendation (Virinchi, 2022); Google develops TensorflowGNN (Sibon Li et al., 2021) for real-time traffic prediction in Google Maps (Oliver Lange, 2020); Meta uses GNNs to improve friend and content recommendations on Facebook and Instagram (MetaAI, 2023); and Alibaba open-sources the AliGraph (Yang, 2019) platform and uses GNNs for fraud detection (Liu et al., 2021b) and risk prediction (Li, 2019). Given these companies' huge investment in labor, time, and resources to develop and deploy GNNs, it is crucial for them to be able to verify the ownership of their own models to protect against illegal copying, model theft, and malicious distribution.

Watermarking, an ownership verification technique, embeds a secret pattern into a model (Uchida et al., 2017) so that if it is stolen or misused, ownership can still be proven through the retained watermark. Among the watermarking strategies (see Section 2), backdoor-based watermarking is the *de facto* approach, especially for non-graph data (Adi et al., 2018; Bansal et al., 2022; Lv et al., 2023; Yan et al., 2023; Li et al., 2022; Shao et al., 2022; Lansari et al., 2023). In these methods, a backdoor trigger (e.g., a logo) is inserted as the watermark pattern into some clean samples (e.g., images) with a *target label* different from the true label, and the model is trained on both the watermarked and clean samples. During verification, ownership is proven by demonstrating that samples with the backdoor trigger consistently produce the target label. Backdoor-based watermarking methods have several merits: they are robust to removal attacks such as model pruning and fine-tuning, and ownership verification only requires black-box access to the target model.

However, recent works (Yan et al., 2023; Liu et al., 2024) show backdoor-based watermarking methods – which are primarily developed for non-graph data – have a fundamental limitation: they induce ownership ambiguity, as attackers could falsely claim misclassified data as ownership evidence. Note that this security issue also exists in the few backdoor-based watermarking methods designed for graph data (Xu et al., 2023). Further, they purposely manipulate the normal model training with *polluted* data samples, which could cause security issues like data poisoning attacks.

Recognizing these limitations, researchers have explored alternate spaces for embedding watermarks. For example, (Shao et al., 2024) embed watermarks into explanations of DNN predictions, avoiding tampering with model predictions or parameters. While (Shao et al., 2024) offers compelling benefits, such as eliminating data pollution risks, their approach assumes a ground-truth watermark is known. This requirement introduces challenges, such as reliance on trusted third parties and potential disputes over the true watermark. Moreover, (Shao et al., 2024) do not address the unique complexities of graph data, including structural dependencies and multi-hop relationships.

Motivated by these insights, we extend explanation-based watermarking to GNNs, addressing the challenges specific to graph data while avoiding reliance on ground-truth watermark verification. Our approach aligns explanations of selected subgraphs with a predefined watermark, ensuring robustness to removal attacks and preserving the advantages of explanation-based methods. In doing so, we present the first explanation-based watermarking method tailored to GNNs.

**Our approach:** We develop a novel watermarking strategy for protecting GNN model ownership that both inherits the merits from and mitigates the drawbacks of backdoor-based watermarking. Like backdoor-based methods, our approach only needs black-box model access. However, in contrast to using *predictions* on the *polluted* watermarked samples, we leverage the *explanations* of GNN predictions on *clean* samples and align them with a predefined watermark for ownership verification. Designing this explanation-based watermarking presents several challenges: First, how do we optimize the GNN training such that these explanations *effectively* align with the watermark? Second, how do we guarantee that this alignment provides *unique* proof of ownership (to eliminate ownership ambiguity)? Third, is the embedded watermark pattern *robust* to removal attacks? And fourth, is the ownership evidence *undetectable* to adversaries?

Addressing these challenges requires careful design. Prior to training, the owner selects a secret set of watermarked subgraphs (private) and defines a watermark pattern (*possibly* private).[1] The GNN is trained with a dual-objective loss function that minimizes (1) standard classification loss, and (2) distance between the watermark and the explanation of each watermarked subgraph. Our method, like GraphLIME (Huang et al., 2023), uses Gaussian kernel matrices to approximate the influence of node features on GNN predictions. However, instead of GraphLIME's iterative approach, we employ ridge regression to compute feature attribution vectors in a single step, providing a more efficient, closed-form solution.

Our approach is (i) *Effective*: We observe that explanations of watermarked subgraphs exhibit high similarity to the watermark after training. (ii) *Unique:* This similarity across explanations is statistically unlikely to be seen in the absence of watermarking, and hence serves as our ownership evidence. (iii) *Undetectable:* We prove that, even with full knowledge of our watermarking method, it is computationally intractable (NP-hard) for adversaries to find the private watermarked subgraphs. (iv) *Robust:* Through empirical evaluations on multiple benchmark graph datasets and GNN models, our method shows robustness to fine-tuning and pruning-based watermark removal attacks. We summarize our contributions as follows:

• We introduce the first known method for watermarking GNNs via their explanations, avoiding data pollution and ownership ambiguity pitfalls in state-of-the-art black-box watermarking schemes.

• We prove that it is NP-hard for the worst-case adversary to identify our watermarking mechanism.

• We show our method is robust to watermark removal attacks like fine-tuning and pruning.

## 2 RELATED WORK

Watermarking techniques can be generally grouped into *white-box* and *black-box* methods.

---

[1]Ownership verification does not use the watermark itself, and will work regardless of whether it is known.

**White-Box Watermarking.** This type of watermarking technique (Darvish Rouhani et al., 2019; Uchida et al., 2017; Wang & Kerschbaum, 2020; Shafieinejad et al., 2021) directly embeds watermarks into the model parameters or features during training. For example, Uchida et al. (2017) propose embedding a watermark in the target model via a regularization term, while Darvish Rouhani et al. (2019) proposed embedding the watermark into the activation/feature maps. Although these methods are robust in theory (Chen et al., 2022), they require full access to the model parameters during verification, which may not be feasible in real-world scenarios, especially for deployed models operating in black-box environments (e.g., APIs).

**Black-Box Watermarking.** Black-box approaches verify model ownership using only model predictions (Adi et al., 2018; Chen et al., 2018; Szyller et al., 2021; Le Merrer et al., 2019). They often use backdoor-based methods, training models to output specific predictions for "trigger" inputs; this owner-specified output can serve as ownership evidence (Adi et al., 2018; Zhang et al., 2018). These methods have significant downsides. First, purposeful data pollution and model manipulation can cause security issues like data poisoning attacks (Steinhardt et al., 2017; Zhang et al., 2019). Further, backdoor-based methods suffer from ambiguity — since they rely on misclassification, attackers may claim naturally-misclassified samples as their own "watermark"(Yan et al., 2023; Liu et al., 2024). Noting these issues with backdoor-based methods, Shao et al. (2024) proposed using explanations as the embedding space for DNN watermarks. This avoids modifying model predictions or parameters, eliminates data pollution risks, and retains compatibility with black-box querying.

**Watermarking GNNs.** There are unique challenges to watermark GNNs—graphs vary widely in size and structure, making it difficult to embed a watermark that can be applied uniformly across different graphs. Moreover, the multi-hop message-passing mechanisms in GNNs are more sensitive to changes in data than other neural networks that process more uniform data, such as images or text (Wang & Gong, 2019; Zügner et al., 2020; Zhou et al., 2023). The only existing black-box method for watermarking GNNs (Xu et al., 2023) is backdoor-based, and suffers from the same data pollution and ownership ambiguity issues as backdoor watermarking of non-graph models (Liu et al., 2024)[2]. These issues, coupled with the complexity of graphs, make existing watermarking techniques unsuitable for GNNs. This highlights the need for novel watermarking approaches.

## 3 BACKGROUND AND PROBLEM FORMULATION

### 3.1 GNNs FOR NODE CLASSIFICATION

Let a graph be denoted as $G = (\mathcal{V}, \mathcal{E}, \mathbf{X})$, where $\mathcal{V}$ is the set of nodes, $\mathcal{E}$ is the set of edges, and $\mathbf{X} = [\mathbf{x}_1, \cdots, \mathbf{x}_N] \in \mathbb{R}^{N \times F}$ is the node feature matrix. $N = |\mathcal{V}|$ is the number of nodes, $F$ is the number of features per node, and $\mathbf{x}_u \in \mathbb{R}^F$ is the node $u$'s feature vector. We assume the task of interest is node classification. In this context, each node $v \in \mathcal{V}$ has a label $y_v$ from a label set $C = \{1, 2, \cdots, C\}$, and we have a set of $|\mathcal{V}^{tr}|$ labeled nodes $(\mathcal{V}^{tr}, \mathbf{y}^{tr}) = \{(v_u^{tr}, y_u^{tr})\}_{u \in \mathcal{V}^{tr}} \subset \mathcal{V} \times C$ nodes as the training set. A GNN for node classification takes as input the graph $G$ and training nodes $\mathcal{V}^{tr}$, and learns a node classifier, denoted as $f$, that predicts the label $\hat{y}_v$ for each node $v$. Suppose a GNN has $L$ layers and a node $v$'s representation in the $l$-th layer is $\mathbf{h}_v^{(l)}$, where $\mathbf{h}_v^{(0)} = \mathbf{x}_v$. Then it updates $\boldsymbol{h}_v^{(l)}$ for each node $v$ using the following two operations:

$$\boldsymbol{l}_v^{(l)} = \text{Agg}(\{\boldsymbol{h}_u^{(l-1)} : u \in \mathcal{N}(v)\}), \; \boldsymbol{h}_v^{(l)} = \text{Comb}(\boldsymbol{h}_v^{(l-1)}, \boldsymbol{l}_v^{(l)}), \quad (1)$$

where Agg iteratively aggregates the representations of all neighbors of a node, and Comb updates the node's representation by combining it with the aggregated neighbors' representations. $\mathcal{N}(v)$ denotes the neighbors of $v$. Different GNNs use different Agg and Comb operations.

The last-layer representation $\mathbf{h}_v^{(L)} \in \mathbb{R}^{|C|}$ of the training nodes $v \in \mathcal{V}^{tr}$ are used for training the node classifier $f$. Let $\Theta$ be the model parameters and $v$'s softmax/confidence scores be $\mathbf{p}_v = f_\Theta(\mathcal{V}^{tr})_v = \text{softmax}(\mathbf{h}_v^{(L)})$, where $p_{v,c}$ indicates the probability of node $v$ being class $c$. Then, $\Theta$ are learned by

---

[2]A recent method, GrOVe (Waheed et al., 2024), is a "fingerprinting" method, verifying ownership of GNNs through node embeddings rather than explicit watermark patterns. However, its authors note it is vulnerable against model pruning attacks. In general, relying on intrinsic model features limits guarantees of uniqueness and can introduce ownership ambiguity (Wang et al., 2021; Liu et al., 2024).

minimizing a classification (e.g., cross-entropy) loss on the training nodes:

$$\Theta^* = \arg \min_{\Theta} \mathcal{L}_{CE}(\mathbf{y}^{tr}, f_{\Theta}(\mathcal{V}^{tr})) = -\Sigma_{v \in \mathcal{V}^{tr}} \ln p_{v,y_v}. \tag{2}$$

## 3.2 GNN Explanation

GNN explanations reveal how a GNN makes decisions by identifying graph features that most influence the prediction. Some methods (e.g., GNNExplainer (Ying et al., 2019) and PGExplainer (Luo et al., 2020)) identify important subgraphs, while others (e.g., GraphLime (Huang et al., 2023)) identify key node features. Inspired by GraphLime (Huang et al., 2023), we use Gaussian kernel matrices to capture relationships between node features and predictions: Gaussian kernel matrices are adept at capturing nonlinear dependencies and complex relationships between variables, ensuring that subtle patterns in the data are effectively represented Yamada et al. (2012). Using these Gaussian kernel matrices, we employ a closed-form solution with ridge regression (Hoerl & Kennard, 1970), allowing us to compute feature importance in a single step.

Our function $explain(\cdot)$ takes node feature matrix $\mathbf{X}$ and nodes' softmax scores $\mathbf{P} = [\mathbf{p}_1, \cdots, \mathbf{p}_N]$, and produces a $F$-dimensional feature attribution vector $\mathbf{e}$, where each entry indicates the positive or negative feature influence on the GNN's predictions across all nodes.

$$\mathbf{e} = explain(\mathbf{X}, \mathbf{P}) = (\tilde{K}^T \tilde{K} + \lambda \mathbf{I}_F)^{-1} \tilde{K}^T \tilde{L} \tag{3}$$

This equation computes feature attributions ($\mathbf{e}$) by leveraging the relationships between input features ($\mathbf{X}$) and output predictions ($\mathbf{P}$) through Gaussian kernel matrices.

We defer precise mathematical definitions to Appendix Section A.2. For high-level understanding, the matrix $\tilde{K}$, of size $N^2 \times F$, encodes pairwise similarities between nodes based on their features, computed using a Gaussian kernel. Similarly, $\tilde{L}$, of size $N^2 \times 1$, uses a Gaussian kernel to encode pairwise similarities between nodes based on their predictions. The term $(\tilde{K}^T \tilde{K} + \lambda \mathbf{I}_F)^{-1}$, where $\lambda$ is a regularization hyperparameter and $\mathbf{I}_F$ is the $F \times F$ identity matrix, solves a ridge regression problem to ensure a stable and interpretable solution. The product $\tilde{K}^T \tilde{L}$, of size $F \times 1$, ties the Gaussian feature similarities ($\tilde{K}$) to the output prediction similarities ($\tilde{L}$), ultimately yielding the vector $\mathbf{e}$, of size $F \times 1$, which quantifies the importance of each input feature for the GNN's predictions.

In this paper, the *explanation* of a GNN's node predictions means this feature attribution vector $\mathbf{e}$.

## 3.3 Problem Formulation

We design an explanation-based watermarking method to protect GNN ownership. This involves defining a watermark pattern (a vector $\mathbf{w}$) and selecting a set of watermarked subgraphs from $G$. Our approach trains a GNN $f$ to embed the relationship between $\mathbf{w}$ and the watermarked subgraphs, enabling the explanations of these subgraphs to serve as verifiable model ownership evidence.

**Threat Model:** There are three parties: the model owner, the adversary, and the third-party model ownership verifier. Obviously, the model owner has white-box access to the target GNN model.

- **Adversary:** We investigate an adversary who dishonestly claims ownership of the GNN model $f$. We primarily assume the adversary does not have direct knowledge of the watermarked subgraphs in $G$. To evaluate the robustness of our method, we allow that the adversary might know *other* details, such as the shape and number of watermarked subgraphs, or the watermark itself. The adversary seeks to undermine the watermarking scheme by (1) attempting to find the watermarked subgraphs (or similarly-convincing alternatives), or (2) implementing a watermark removal attack.

- **Model Ownership Verifier:** Following existing backdoor-based watermarking, we use black-box ownership verification, where the verifier does not need full access to the protected model.

**Objectives:** Our explanation-based watermarking method aims to achieve the below objectives:

1. **Effectiveness.** Training must embed the watermark in the explanations of our selected subgraphs: their feature attribution vectors must be *sufficiently*[3] aligned with vector $\mathbf{w}$.

---

[3] Note: alignment between explanations and $\mathbf{w}$ is a tool for the owner to measure optimization success; for a watermark to function as ownership evidence, alignment must simply be "good enough" (See Section 5.2.1).

Figure 1: Overview of our explanation-based GNN watermarking method. During embedding, $f$ is optimized to (1) minimize node classification loss and (2) align explanations of watermarked subgraphs with $\mathbf{w}$. During ownership verification, the similarity of $G^{cdt}$'s binarized explanations, $\{\hat{\mathbf{e}}_i^{cdt}\}_{i=1}^T$, is tested for significance. In this example, $G^{cdt}$ are *not* the watermarked subgraphs; as a result, $\{\hat{\mathbf{e}}_i^{cdt}\}_{i=1}^T$ fail to exhibit significant similarity and are rejected.

2. **Uniqueness.** Aligning watermarked subgraph explanations with $\mathbf{w}$ must yield statistically-significant similarity between explanations that is unlikely to occur in alternate solutions.

3. **Robustness.** The watermark must be robust to removal attacks like fine-tuning and pruning.

4. **Undetectability.** Non-owners should be unable to locate the watermarked explanations.

## 4 METHODOLOGY

Our watermarking method occurs in three stages: (1) design, (2) embedding, and (3) ownership verification. Since design relies on embedding and ownership verification requirements, we introduce stages (2) and (3) beforehand. Training $f$ involves a dual-objective loss function balancing node classification and watermark embedding. Minimizing watermark loss reduces the misalignment between $\mathbf{w}$ and the explanations of $f$'s predictions on the watermarked subgraphs, embedding the watermark. During ownership verification, explanations are tested for statistically-significant similarity due to their common alignment with $\mathbf{w}$. Lastly, we detail watermark design principles, which ensure the similarity observed across our explanations is statistically-significant, unambiguous ownership evidence. Figure 1 gives an overview of our explanation-based watermarking method.

### 4.1 WATERMARK EMBEDDING

Let training set $\mathcal{V}^{tr}$ be split as two disjoint subsets: $\mathcal{V}^{clf}$ for node classification and $\mathcal{V}^{wmk}$ for watermarking. Select $T$ subgraphs $\{G_1^{wmk}, \ldots, G_T^{wmk}\}$ whose nodes $\{\mathcal{V}_i^{wmk}\}_{i=1}^T$ will be watermarked. These subgraphs have explanations $\{\mathbf{e}_1^{wmk}, \ldots, \mathbf{e}_T^{wmk}\}$, where $\mathbf{e}_i^{wmk} = explain(\mathbf{X}_i^{wmk}, \mathbf{P}_i^{wmk})$ explains $f$'s softmax output $\mathbf{P}_i^{wmk}$ on $G_i^{wmk}$'s nodes $\mathcal{V}_i^{wmk}$, which have features $\mathbf{X}_i^{wmk}$. Define watermark $\mathbf{w}$ as an $M$-dimensional vector ($M \leq F$), whose entries are 1s and $-1$s.

Inspired by Shao et al. (2024), we use multi-objective optimization to balance classification performance with a hinge-like watermark loss function. When minimized, the *watermark loss* encourages alignment between $\mathbf{w}$ and $\{\mathbf{e}_i^{wmk}\}_{i=1}^T$, embedding the relationship between $\mathbf{w}$ and these subgraphs.

$$\mathcal{L}_{wmk}(\{\mathbf{e}_i^{wmk}\}_{i=1}^T, \mathbf{w}) = \sum_{i=1}^T \sum_{j=1}^M \max(0, \epsilon - \mathbf{w}[j] \cdot \mathbf{e}_i^{wmk}[\mathbf{idx}[j]]), \qquad (4)$$

where $\mathbf{e}_i^{wmk}[\mathbf{idx}]$ represents the *watermarked portion* of $\mathbf{e}_i^{wmk}$ on node feature indices $\mathbf{idx}$ with length $M$; $\mathbf{idx}$ is same for all explanations $\{\mathbf{e}_i^{wmk}\}_{i=1}^T$. We emphasize that $\mathbf{idx}$ are not arbitrary, but are rather the result of design choices discussed later in Section 4.3. The hyperparameter $\epsilon$ bounds the contribution of each multiplied pair $\mathbf{w}[j] \cdot \mathbf{e}_i^{wmk}[\mathbf{idx}[j]]$ to the summation.

We train the GNN model $f$ to minimize both classification loss on the nodes $\mathcal{V}^{clf}$ (see Equation 2) and watermark loss on the explanations of $\{G_1^{wmk}, \ldots, G_T^{wmk}\}$, with a balancing hyperparameter $r$:

$$\min_{\Theta} \mathcal{L}_{CE}(\mathbf{y}^{clf}, f_\Theta(\mathcal{V}^{clf})) + r \cdot \mathcal{L}_{wmk}(\{\mathbf{e}_i^{wmk}\}_{i=1}^T, \mathbf{w}) \qquad (5)$$

After training, we expect the learned parameters $\Theta$ to ensure not only an accurate node classifier, but also similarity between $\mathbf{w}$ and explanations $\{\mathbf{e}_i^{wmk}\}_{i=1}^T$ at indices $\mathbf{idx}$.

Algorithm 1 (in Appendix) provides a detailed description.

## 4.2 OWNERSHIP VERIFICATION

Since they were aligned with the same $\mathbf{w}$, explanations $\{\mathbf{e}_i^{cdt}\}_{i=1}^T$ will be similar to each other after training. Therefore, when presented with $T$ *candidate subgraphs* $\{\mathbf{e}_1^{cdt}, \mathbf{e}_2^{cdt}, \cdots, \mathbf{e}_T^{cdt}\}$ by a purported owner (note that our threat model assumes a strong adversary who also knows $T$), we must measure the similarity between these explanations to verify ownership. If the similarity is statistically significant at a certain level, we can conclude the purported owner knows which subgraphs were watermarked during training, and therefore that they are the true owner.

**Explanation Matching:** Our GNN explainer in Equation (3) produces a positive or negative score for each node feature, indicating its influence on the GNN's predictions, generalized across all nodes in the graph. To easily compare these values across candidate explanations, we first *binarize* them with the sign function. For the $j^{th}$ index of an explanation $\mathbf{e}_i^{cdt}$, this process is defined as:

$$\hat{\mathbf{e}}_i^{cdt}[j] = \begin{cases} 1 & \text{if } \mathbf{e}_i^{cdt}[j] > 0 \\ -1 & \text{if } \mathbf{e}_i^{cdt}[j] < 0 \\ 0 & \text{otherwise} \end{cases} \tag{6}$$

We then count the *matching indices* (MI) across all the binarized explanations — the number of indices at which all binarized explanations have matching, non-zero values:[4]

$$\text{MI}^{cdt} = \text{MI}(\{\hat{\mathbf{e}}_i^{cdt}\}_{i=1}^T) = \Sigma_{j=1}^F \mathbb{1}((\{\hat{\mathbf{e}}_i^{cdt}[j] \neq 0, \forall i\}) \wedge (\hat{\mathbf{e}}_1^{cdt}[j] = \hat{\mathbf{e}}_2^{cdt}[j] = \cdots = \hat{\mathbf{e}}_T^{cdt}[j])) \tag{7}$$

**Approximating a Baseline MI Distribution:** To test the significance of $\text{MI}^{cdt}$, we need to approximate the distribution of *naturally-occurring* matches: the MIs for all $T$-sized sets of un-watermarked explanations. We perform $I$ (which should be sufficiently large; $I = 1000$ in our experiments) simulations by randomly sampling sets of $T$ subgraphs from the training graph, and obtaining the MI of the binarized explanations of each set of the subgraphs. Then, we can obtain *empirical* estimates of mean and standard deviation, $\mu_{nat_e}$ and $\sigma_{nat_e}$ (note the subscript "e"), for these $I$ MIs.

**Significance Testing to Verify Ownership:** We verify the purported owner's ownership by testing if $\text{MI}^{cdt}$ is statistically unlikely for randomly selected subgraphs, at some significance level $\alpha_v$:

$$Ownership = \begin{cases} True & \text{if } p_{z_{test}} < \alpha_v \\ False & \text{otherwise} \end{cases} \quad \text{where} \quad z_{test} = \frac{\text{MI}^{cdt} - \mu_{nat_e}}{\sigma_{nat_e}} \tag{8}$$

Algorithm 2 (in Appendix) provides a detailed description of the ownership verification process.

## 4.3 WATERMARK DESIGN

The watermark $\mathbf{w}$ is an $M$-dimensional vector with entries of 1 and $-1$. The size and location of $\mathbf{w}$ must allow us to *effectively* embed *unique* ownership evidence into our GNN.

**Design Goal:** The watermark should be designed to yield a *target MI* ($\text{MI}^{tgt}$) that passes the statistical test in Equation (8). This value is essentially the upper bound on a one-sided confidence interval. However, since we cannot obtain the estimates $\mu_{nat_e}$ or $\sigma_{nat_e}$ without a trained model, we instead use a binomial distribution to *predict* estimates $\mu_{nat_p}$ and $\sigma_{nat_p}$ (note the subscript "p").

We assume the random case, where a binarized explanation includes values $-1$ or 1 with equal probability (again, ignoring zeros; see Footnote 4). Across $T$ binarized explanations, the probability of a match at an index is $p_{match} = 2 \times 0.5^T$. We estimate $\mu_{nat_p} = F \times p_{match}$ (where $F$ is number of node features), and $\sigma_{nat_p} = \sqrt{F \times p_{match}(1 - p_{match})}$. We therefore define $\text{MI}^{tgt}$ as follows:

$$\text{MI}^{tgt} = min(\mu_{nat_p} + \sigma_{nat_p} \times z_{tgt}, F), \tag{9}$$

---

[4]We exclude 0s from our count of MI because a 0 in the explanation corresponds to 0 dependence between a node feature and the GNN's prediction, and it is highly unlikely for the optimization process to achieve this level of precision unless the explanation index corresponds to a node feature with zero value. Therefore, we conclude all 0's must reflect naturally occurring zeros in $\mathbf{X}$ and are irrelevant to measurements of watermarking.

where $z_{tgt}$ is the $z$-score associated with target significance $\alpha_{tgt}$. In practice, we set $\alpha_{tgt} = 1e-5$; since $\text{MI}^{tgt}$ affects watermark design, we want to ensure it does not underestimate the upper bound.

However, two questions remain: 1) What watermark size $M$ will allow us to reach an $\text{MI}^{tgt}$, and 2) which indices **idx** should be watermarked with these $M$ values?

**Watermark Length $M$:** For $T$ binarized explanations, our estimated lower bound of baseline MI is:

$$\text{MI}^{LB} = max(\mu_{nat_p} - \sigma_{nat_p} \times z_{LB}, 0), \tag{10}$$

where $z_{LB}$ is the $z$-score for target significance, $\alpha_{LB}$ — in practice, $\alpha_{LB}$ equals $\alpha_{tgt}$ ($1e-5$).

We expect that at most, our watermark needs to add $(\text{MI}^{tgt} - \text{MI}^{LB})$ net MI. However, if some indices in the $T$ binarized explanations already match naturally, the watermark may not add additional net matches. We pad the watermark length to reflect this, so the number of watermarked indices does not fail to contribute a sufficient number of new MI. We calculate the padding based on the probability of a match existing naturally without watermarking. In the most challenging scenario, where $\text{MI}^{tgt}$ MI occurs naturally, the probability of a watermarked index producing a new match is $(F - \text{MI}^{tgt})/F$. Consequently, we pad the required $M$ by the inverse of this probability, $F/(F - \text{MI}^{tgt})$:

$$M = \lceil (\text{MI}^{tgt} - \text{MI}^{LB}) \times F/(F - \text{MI}^{tgt}) \rceil \tag{11}$$

Using watermark length $M$ should yield enough net MI to reach the total, $\text{MI}^{tgt}$, that the owner will need to demonstrate ownership. Notice that, under the assumption that we set $\alpha_{LB}$ equal to $\alpha_{tgt}$, Equation (11) is ultimately a function of three variables: $\alpha_{tgt}$, $F$, and $T$.

**Watermark Location idx:** Each explanation corresponds to node feature indices. It is easiest to watermark indices where features are non-zero. We advise selecting **idx** from the $M$ most frequently non-zero node features across all $T$ watermarked subgraphs. Let $\mathbf{X}^{wmk} = [\mathbf{X}_1^{wmk}; \mathbf{X}_2^{wmk}; \cdots \mathbf{X}_T^{wmk}]$ be the concatenation of node features of the $T$ watermarked subgraphs. Then, we define **idx** as:

$$\mathbf{idx} = \text{top}_M \left( \{ \|\mathbf{x}_1^{wmk}\|_0, \|\mathbf{x}_2^{wmk}\|_0, \cdots, \|\mathbf{x}_F^{wmk}\|_0 \} \right), \tag{12}$$

where $\mathbf{x}_j^{wmk}$ is the $j$-th column of $\mathbf{X}^{wmk}$, $\| \cdot \|_0$ represents the number of non-zero entries in a vector, and $\text{top}_M(\cdot)$ returns the indices of the $M$ largest values.

### 4.4 Locating the Watermarked Subgraphs

An adversary may search for the watermarked subgraphs to falsely claim ownership. In the worst case, they will have access to $G^{tr}$ and know both the number of watermarked subgraphs $T$, and the node size $s$ of each subgraph. With $G^{tr}$, the adversary can compute the distribution $(\mu_{nat_e}, \sigma_{nat_e})$ of naturally-occurring matches, and then search for $T$ subgraphs whose binarized explanations have maximally-significant MI. They can do this in two ways: a brute-force search or a random search.

**Brute-Force Search:** If the training graph has $N$ nodes, identifying $n_{sub} = sN$-node subgraphs yields $\binom{N}{n_{sub}}$ options. To find the $T$ subgraphs with a maximum MI across their binarized explanations, an adversary must compare all $T$-sized sets of these subgraphs, with $\binom{\binom{N}{n_{sub}}}{T}$ sets in total.

**Random Search:** Alternatively, an adversary can randomly sample subgraphs in the hopes of finding a group that is "good enough". To do this, they make $T$ random selections of an $n_{sub}$-sized set of nodes, each of which comprises a subgraph. Given $N$ training nodes and $T$ watermarked subgraphs of size $n_{sub}$, the probability that an attacker-chosen subgraph of size $n_{sub}$ overlaps with any single watermarked subgraph with no less than $j$ nodes is given as:

$$P(\text{at least } j \text{ overlapping nodes}) = 1 - \left( \sum_{m=1}^{j} \binom{n_{sub}}{m} \binom{N - n_{sub}}{n_{\text{sub}} - m} \middle/ \binom{N}{n_{\text{sub}}} \right)^T \tag{13}$$

The summation represents the probability that a randomly selected subgraph contains less than $j$ nodes from a watermarked subgraph. Raising this to the power of $T$ yields the probability that overlap $< j$ for all watermarked subgraphs. Subtracting this from 1 yields the probability that the randomly selected subgraph contains at least $j$ nodes from the same watermarked subgraph.

In Section 5.2.3 we demonstrate the infeasibility of both brute-force and random search.

| Dataset | GCN | | | | SGC | | | | SAGE | | | |
|---|---|---|---|---|---|---|---|---|---|---|---|---|
| | Accuracy (Trn/Tst) no wmk \| wmk | | Wmk Alignmt | MI $p$-val | Accuracy (Trn/Tst) no wmk \| wmk | | Wmk Alignmt | MI $p$-val | Accuracy (Trn/Tst) no wmk \| wmk | | Wmk Alignmt | MI $p$-val |
| Photo | 91.3/89.4\| 90.9/88.3 | | 91.4 | <0.001 | 91.4/89.9\| 90.1/88.0 | | 91.8 | <0.001 | 94.2/90.8\| 94.1/88.2 | | 97.7 | <0.001 |
| PubMed | 88.6/85.8\| 85.7/81.4 | | 91.5 | <0.001 | 88.8/85.9\| 85.3/81.4 | | 88.9 | <0.001 | 90.5/86.0\| 91.1/81.2 | | 85.2 | <0.001 |
| CS | 98.5/90.3\| 96.8/89.8 | | 73.8 | <0.001 | 98.4/90.3\| 96.7/90.1 | | 74.5 | <0.001 | 100./88.4\| 99.9/88.9 | | 78.2 | <0.001 |

Table 1: Watermarking results. Each value is the average of five trials with distinct random seeds. Subscripts *w* and *n* indicate results from training with and without the watermark, respectively.

## 5 EXPERIMENTS

### 5.1 SETUP

**Datasets and Training/Testing Sets:** We evaluate our watermarking method on three standard datasets commonly used in node classification tasks: Amazon Photo — a subset of the Amazon co-purchase network (McAuley et al., 2015), CoAuthor CS — a coauthorship network (Shchur et al., 2019), and PubMed — a citation network (Yang et al., 2016). (See Appendix A.1 for more details.)

The graph is split into three sets: 60% nodes for training, 20% for testing, and the remaining 20% for further training tasks, such as fine-tuning or other robustness evaluations. As mentioned in Section 4.1, training nodes are further split into two disjoint sets: one for training the GNN classifier, and one consisting of the watermarked subgraphs. (Their relative sizes are determined by the size and number of watermarked subgraphs, which are hyperparameters mentioned below.) The test set is used to evaluate classification performance after training. The remaining set enables additional training of the pre-trained GNN on unseen data to assess watermark robustness.

**GNN Models and Hyperparameters:** We apply our watermarking method to three GNN models: GCN Kipf & Welling (2017), SGC (Wu et al., 2019), and GraphSAGE (Hamilton et al., 2018). Our main results use the GraphSAGE architecture by default. Unless otherwise specified, we use $T = 4$ watermarked subgraphs, each with the size $s = 0.5\%$ of the training nodes. Key hyperparameters in our watermarking method, including the significance levels ($\alpha_{tgt}$ and $\alpha_v$), balanced hyperparameter ($r$), and watermark loss contribution bound ($\epsilon$), were tuned to balance classification and watermark losses. A list of all hyperparameter values are in the Appendix. *Note that our watermark design in Equation (11) allows us to learn the watermark length M.*

### 5.2 RESULTS

As stated in Section 3.3, successful watermark should exhibit effectiveness, uniqueness, robustness, and undetectability. Our experiments aim to assess each of these. More results see Appendix.

#### 5.2.1 EFFECTIVENESS AND UNIQUENESS

Embedding *effectiveness* can be measured by the alignment of the binarized explanations with the watermark pattern **w** at indices **idx**; this metric can be used by the owner to confirm that **w** was effectively embedded in $f$ during training. Since the entries of **w** are 1s and −1s, we simply count the average number of watermarked indices at which a binarized explanation matches **w**:

$$Watermark\ Alignment = (1/T) \times \Sigma_{i=1}^{T} \Sigma_{j=1}^{M} \mathbb{1}(\hat{\mathbf{e}}_i^{wmk}[\mathbf{idx}[j]] = \mathbf{w}[j]) \qquad (14)$$

Watermarking *uniqueness* is measured by the MI $p$-value for the binarized explanations of the $T$ watermarked subgraphs, as defined by Equation (8). A low $p$-value indicates that the MI of the watermarked explanations is statistically unlikely to observed in explanations of randomly selected subgraphs. *This metric is more important than watermark alignment*; as long as the watermarked subgraphs yield a uniquely large MI, it is sufficient, even if alignment is under 100%.

Table 1 shows results under the default setting, averaged over five trials with distinct random seeds and watermark patterns. It highlights our method's *effectiveness*, *uniqueness*, and classification performance. The key result is the MI $p$-value, which shows *uniqueness* of the ownership claim; this remains below 0.001 in all cases where $T > 2$, even when watermark alignment is below 100%. Accuracy remains high across datasets and models, showing minimal impact from watermarking.

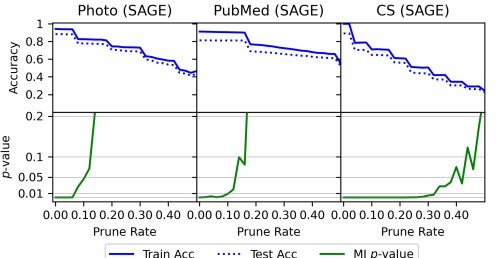 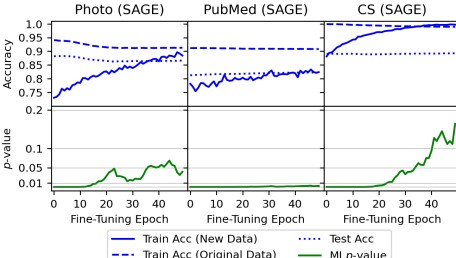

Figure 2: Effect of pruning (left) and fine-tuning (right) on MI $p$-value. These results reflect our default architecture (GraphSAGE), number of subgraphs ($T$), and subgraph size ($s$). See Appendix for results for varied architectures, $T$, $s$, and learning rates (Figures 6- 9, 10, 11, and 12, respectively.)

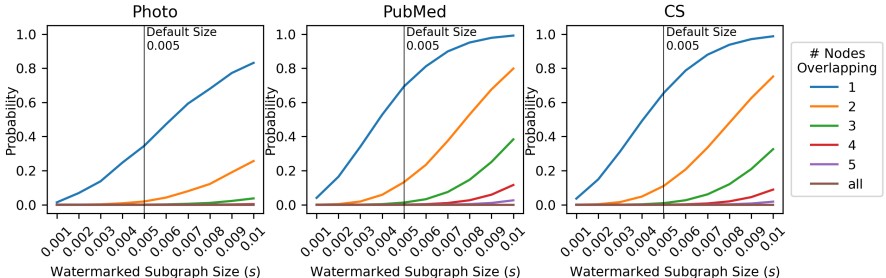

Figure 3: The probability that a randomly-chosen subgraph overlaps with a watermarked subgraph.

### 5.2.2 ROBUSTNESS

A good watermark will be robust to removal attacks. We explore two types of attacks. The first is pruning for model compression (Li et al., 2016), as used to assess watermark robustness by Liu et al. (2021a), Tekgul et al. (2021), and others. The second is fine-tuning Pan & Yang (2010), as used for robustness assessment by Adi et al. (2018), Wang et al. (2020), and more.

**Pruning:** Pruning is a model compression strategy that sets a portion of weights to zero (Li et al., 2016). The particular approach we explore, *structured* pruning, targets rows and columns of parameter tensors — such as node embeddings and edge features — based on their importance scores, or $L_n$-norms (Paszke et al., 2019). An attacker hopes that by pruning the model, they may remove the watermark while still maintaining high classification accuracy.

**Fine-Tuning:** Fine-tuning is a technique that continues training on previously trained models to adapt to a new task (Pan & Yang, 2010). An attacker may use fine-tuning to get the GNN to "forget" the watermark. To test our model's robustness to this type of attack, we continue training the model on the *validation* dataset, $G^{val}$, at 0.1 times the original learning rate for 49 epochs. (See Appendix Section A.5 for results with other learning rates and GNN architectures.)

Figure 2 shows the impact of pruning and fine-tuning attacks. The left shows the impact of pruning rates 0.0 (no GNN parameters pruned) to 1.0 (all pruned). In all datasets, the MI $p$-value only rises as classification accuracy drops, meaning the owner would notice before the pruning affects the watermark. The right shows classification accuracy and MI $p$-value in a fine-tuning attack. CS has a near-zero MI $p$-value for about 25 epochs, whereas Photo and PubMed have low MI $p$-values for the full duration. This demonstrates the watermark's robustness for extended periods during fine-tuning.

### 5.2.3 UNDETECTABILITY

**Brute-Force Search:** With Equations from Section 4.4, we use our smallest datset, Amazon Photo (4590 training nodes), to demonstrate the infeasibility of a brute-force search for the watermarked subgraphs. We assume adversaries know the number ($T$) and size ($s$) of our watermarked subgraphs. With default $s = 0.005$, each watermarked subgraph has $ceil(0.005 \times 4590) = 23$ nodes — there are $\binom{4590}{23} = 6.1 \times 10^{61}$ subgraphs of this size; with default $T = 4$, there are $\binom{\binom{4590}{23}}{4} = 5.8 \times 10^{245}$ possible $T$-sized sets of candidate subgraphs. Therefore, even in our smallest dataset, finding the *uniquely-convincing* set of watermarked subgraphs is an incredibly hard problem.

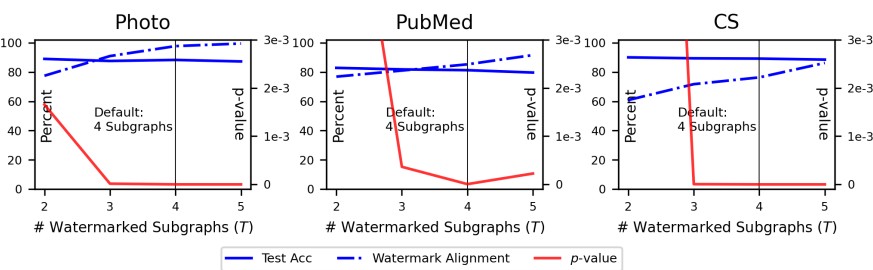

Figure 4: Watermarking metrics for varied number of watermarked subgraphs, $T$.

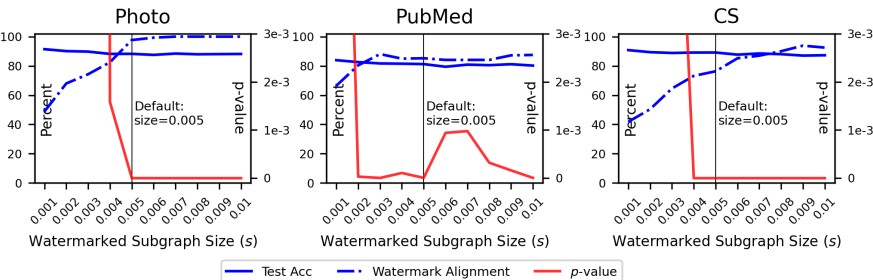

Figure 5: Watermarking metrics for varied watermarked subgraph size, $s$.

**Random Search:** Figure 3 shows the probability (from Equation 13), for varied subgraph sizes $s$, that $j$ nodes of a randomly-chosen subgraph overlap with any single watermarked subgraph. The figure plots these values for $j = 1, 2, 3, 4, 5$, or all $n_{sub}$ watermarked subgraph nodes and our default $T = 4$ watermarked subgraphs. For our default subgraph size of $s = 0.005$ (or equivalently, 0.5% of the training nodes), there is close to 0 probability that a randomly-selected subgraph will contain 3 or more nodes that overlap with a common watermarked subgraph. This demonstrates very low probability that a randomly-selected subgraph will be similar to the actual watermarked subgraphs.

## 5.3 Ablation Studies

In this section, we explore the role of (1) watermarked subgraph size and (2) the number of watermarked subgraphs on the effectiveness, uniqueness, and robustness of the watermark.

**Impact of the Number of Watermarked Subgraphs $T$:** Figure 4 shows how the number of watermarked subgraphs, $T$, affects various watermark performance metrics. The results show that for all datasets, larger $T$ increases watermark alignment and a lower $p$-value, although test accuracy decreases slightly for Photo and PubMed datasets. Notably, our default of $T = 4$ is associated with a near-zero $p$-value in every scenario. Figure 10 in Appendix also shows the robustness results to removal attacks against varied $T$: we observe that the watermarking method resists pruning attacks until test accuracy is affected, and fine-tuning attacks for at least 25 epochs for any dataset.

**Impact of the Size of Watermarked Subgraphs $s$:** Figure 5 shows the results with different sizes $s$ of the watermarked subgraphs. We observe similar trends as Figure 4: watermarking is generally more effective, unique, and robust for larger values of $s$. Again, we observe a trade-off between subgraph size and test accuracy, though this trend is slight. We note that for $s \geq 0.003$, our method achieves near-zero $p$-values across all datasets, as well as increasing watermark alignment. Figure 11 in Appendix shows the robustness results: across all datasets, when $s > 0.005$, our method is robust against pruning attacks generally, and against fine-tuning attacks for at least 25 epochs.

## 6 Conclusion

In this paper, we introduce the first-known method for watermarking GNNs via their explanations. This avoids common pitfalls of backdoor-based methods: our watermark is designed with a statistical guarantee of unambiguity, and since it does not reside within the training data space, it is not vulnerable to attacks on the data itself. We demonstrate the robustness of our method to removal attacks, while also highlighting the statistical infeasibility of locating the watermarked subgraphs. This presents a significant step forward in securing GNNs against intellectual property theft.

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

## A APPENDIX

---

**Algorithm 1:** Watermark Embedding

---

**Input:** Graph $G$, training nodes $\mathcal{V}^{tr}$, learning rate $\eta$, #watermarked subgraphs $T$, watermarked
   subgraph size $s$, hyperparameter $r$, target significance $\alpha_{tgt}$, watermark loss contribution bound $\epsilon$.
**Output:** A trained and watermarked model, $f$.
**Setup:** Initialize $f$ and optimizer. With $\alpha_{tgt}$, $T$, and number of node features $F$ as input, compute
   $M$ using equation 11. Initialize $\mathbf{w}$ with values $1$ and $-1$ uniform at random. With
   $n_{sub} = ceil(s \times |\mathcal{V}^{tr}|)$, randomly sample $T$ sets of $n_{sub}$ nodes from $\mathcal{V}^{tr}$. These subgraphs jointly
   comprise $G^{wmk}$. Define node set $\mathcal{V}^{clf}$ for classification from the remaining nodes in $\mathcal{V}^{tr}$.
**for** epoch=1 to #Epoch **do**
   $\quad L^{clf} \leftarrow \mathcal{L}_{CE}(\mathbf{y}^{clf}, f_\Theta(\mathcal{V}^{clf}))$
   $\quad L^{wmk} \leftarrow 0$
   $\quad$**for** i=1 to T **do**
   $\quad\quad \mathbf{P}_i^{wmk} \leftarrow f_\Theta(\mathcal{V}_i^{wmk})$
   $\quad\quad \mathbf{e}_i^{wmk} \leftarrow explain(\mathbf{X}_i^{wmk}, \mathbf{P}_i^{wmk})$
   $\quad\quad L^{wmk} \leftarrow L^{wmk} + \sum_{j=1}^M \max(0, \epsilon - \mathbf{w}[j] \cdot \mathbf{e}_i^{wmk}[\mathbf{idx}[j]])$
   $\quad L \leftarrow L^{clf} + r \cdot L^{wmk}$
   $\quad \Theta \leftarrow \Theta - \eta \frac{\partial L}{\partial \Theta}$

---

### A.1 EXPERIMENTAL SETUP DETAILS

**Hardware and Software Specifications.** All experiments were conducted on a MacBook Pro
(Model Identifier: MacBookPro18,3; Model Number: MKGR3LL/A) with an Apple M1 Pro chip (8
cores: 6 performance, 2 efficiency) and 16 GB of memory, on macOS Sonoma Version 14.5. Models
were implemented in Python with the PyTorch framework.

**Dataset Details.** Amazon Photo (simply "Photo" in this paper) is a subset of the Amazon co-
purchase network (McAuley et al., 2015). Nodes are products, edges connect items often purchased
together, node features are bag-of-words product reviews, and class labels are product categories.
Photo has 7,650 nodes, 238,163 edges, 745 node features, and 8 classes. The CoAuthor CS dataset
("CS" in this paper) (Shchur et al., 2019) is a graph whose nodes are authors, edges are coauthorship,
node features are keywords, and class labels are the most active fields of study by those authors. CS
has 18,333 nodes, 163,788 edges, 6,805 node features, and 15 classes. Lastly, PubMed (Yang et al.,
2016) is a citation network whose nodes are documents, edges are citation links, node features are
TF-IDF weighted word vectors based on the abstracts of the papers, and class labels are research
fields. The graph has 19,717 nodes, 88,648 edges, 500 features, and 3 classes.

**Hyperparameter Setting Details.**

Classification training hyperparameters:

- Learning rate: 0.001-0.001
- Number of layers: 3
- Hidden Dimensions: 256-512
- Epochs: 100-300

Watermarking hyperparameters:

- Target significance level, $\alpha_{tgt}$: set to 1e-5 to ensure a watermark size that is sufficiently large.
- Verification significance level, $\alpha_v$: set to 0.01 to limit false verifications to under 1% likelihood.
- Watermark loss coefficient, $r$: set to values between 20-100, depending on the amount required to
  bring $L^{wmk}$ to a similar scale as $L^{clf}$ to ensure balanced learning.
- Watermark loss parameter $\epsilon$: set to values ranging from 0.01 to 0.1. Smaller values ensure that no
  watermarked node feature index has undue influence on watermark loss.

---

**Algorithm 2:** Ownership Verification

---

**Input:** A GNN $f$ trained by Alg. 1, a graph $G$ with training nodes $\mathcal{V}^{tr}$, a collection of $T$ *candidate subgraphs* with node size $n_{sub}$, and a significance level $\alpha_v$ required for verification, $I$ iterations.

**Output:** Ownership verdict.

---

*Phase 1 – Obtain distribution of naturally-occurring matches*

---

**Setup:**
1. Define subgraphs $\mathcal{S} = \{G_1^{rand}, \cdots, G_D^{rand}\}$, where each subgraph is size $n_{sub} = ceil(s \times |\mathcal{V}^{tr}|)$. Each subgraph $G_i^{rand}$ is defined by randomly selecting $n_{sub}$ nodes from $\mathcal{V}^{tr}$. $D$ should be "sufficiently large" ($D > 100$) to approximate a population.

2. Using Equation 6, collect *binarized explanations*, $\hat{\mathbf{e}}_i^{rand}$, for $1 \le i \le D$.

3. Initialize empty list, $matchCounts = \{\}$.

**for** i=1 to I simulations **do**

> Randomly select $T$ distinct indices $idx_1, \ldots, idx_T$ from the range $\{1, \cdots, D\}$.
> For each $idx_i$, let $\mathcal{V}_{idx_i}^{rand}$ and $\mathbf{X}_{idx_i}^{rand}$ be the nodes of $G_{idx_i}^{rand}$ and their features, respectively.
> Compute $\hat{\mathbf{e}}_{idx_i}^{rand} = sign(explain(\mathbf{X}_{idx_i}^{rand}, f(\mathcal{V}_{idx_i}^{rand})))$ for each $i$ in $1 \le i \le T$.
> Compute the MI on $\{\hat{\mathbf{e}}_{idx_1}^{rand}, \cdots, \hat{\mathbf{e}}_{idx_T}^{rand}\}$ using Equation 7, and append to $matchCounts$.

Compute $\mu_{nat_e} = \frac{\Sigma_{i=1}^{I} matchCounts[i]}{I}$ and $\sigma_{nat_e} = \sqrt{\frac{1}{I}\Sigma_{i=1}^{I}(matchCounts[i] - \mu_{nat_e})^2}$.

---

*Phase 2 – Significance testing*

---

Consider the null hypothesis, $H_0$, that the observed MI across $T$ binarized explanations in $\{\hat{\mathbf{e}}_i^{cdt}\}_{i=1}^{T}$ comes from the population of naturally-occurring matches. We conduct a *z*-test to test $H_0$:

1. For $1 \le i \le T$, let $\mathbf{P}_i^{cdt} = f(\mathcal{V}_i^{cdt})$ and $\mathbf{X}_i^{cdt}$ be the corresponding features of $\mathcal{V}_i^{cdt}$.

2. Let the binarized explanation of the $i^{th}$ candidate subgraph be defined as:

$$\hat{\mathbf{e}}_i^{cdt} = sign\left(explain(\mathbf{X}_i^{cdt}, \mathbf{P}_i^{cdt})\right)$$

3. Compute $\text{MI}^{cdt}$ across tensors in $\{\hat{\mathbf{e}}_i^{cdt}\}_{i=1}^{T}$ using Equation 14.

4. Compute the significance of this value as the p-value of a one-tailed *z*-test:

$$z_{test} = \frac{\text{MI}^{cdt} - \mu_{nat_e}}{\sigma_{nat_e}} \quad p_{z_{test}} = 1 - \Phi(z_{test}),$$

Where $\Phi(z_{test})$ is the cumulative distribution function of the standard normal distribution.

5. If $p_{z_{test}} \ge \alpha_v$, the candidate subgraphs *do not* provide adequate ownership evidence. If $p_{z_{test}} < \alpha_v$, the candidate subgraphs provide enough evidence of ownership to reject $H_0$.

---

## A.2 GAUSSIAN KERNEL MATRICES

Define $\bar{\mathbf{K}}$ as a collection of matrices $\{\bar{\mathbf{K}}^{(1)}, \ldots, \bar{\mathbf{K}}^{(F)}\}$, where $\bar{\mathbf{K}}^{(k)}$ (size $N \times N$) is the centered and normalized version of Gaussian kernel matrix $\mathbf{K}^{(k)}$, and each element $\mathbf{K}_{uv}^{(k)}$ is the output of the Gaussian kernel function on the $k^{th}$ node feature for nodes $u$ and $v$:

$$\bar{\mathbf{K}}^{(k)} = \mathbf{H}\mathbf{K}^{(k)}\mathbf{H}/\|\mathbf{H}\mathbf{K}^{(k)}\mathbf{H}\|_F, \ \mathbf{H} = \mathbf{I}_N - \frac{1}{N}\mathbf{1}_N\mathbf{1}_N^T, \ \mathbf{K}_{uv}^{(k)} = \exp\left(-\frac{1}{2\sigma_x^2}\left(\mathbf{x}_u^{(k)} - \mathbf{x}_v^{(k)}\right)^2\right) \quad (15)$$

$\|\cdot\|_F$ is the Frobenius norm, $\mathbf{H}$ is a centering matrix (where $\mathbf{I}_N$ is an $N \times N$ identity matrix and $\mathbf{1}_N$ is an all-one vector of length $N$), and $\sigma_x$ is Gaussian kernel width. Now take the nodes' softmax scores $\mathbf{P} = [\mathbf{p}_1, \cdots, \mathbf{p}_N]$, and their Guassian kernel width, $\sigma_{\mathbf{p}}$. Define $\bar{\mathbf{L}}$ as a centered and normalized $N \times N$

---

Gaussian kernel $L$, where $L_{uv}$ is the similarity between nodes $u$ and $v$'s softmax outputs:

$$\bar{L} = HLH/\|HLH\|_F, \quad L_{uv} = \exp\left(-\frac{1}{2\sigma_{\mathbf{p}}^2}\|\mathbf{p}_u - \mathbf{p}_v\|_2^2\right) \quad (16)$$

Let $\tilde{K}$ be the $N^2 \times F$ matrix $[\text{vec}(\bar{K}^{(1)}), \ldots, \text{vec}(\bar{K}^{(F)})]$, where $\text{vec}(\cdot)$ converts each $N \times N$ matrix $\bar{K}^{(k)}$ into a $N^2$-dimensional column vector. Similarly, we denote $\tilde{L} = \text{vec}(\bar{L})$ as the $N^2$-dimensional, vector form of the matrix $\bar{L}$. Also take $F \times F$ identity matrix $I_F$ and regularization hyperparameter $\lambda$.

### A.3 TIME COMPLEXITY ANALYSIS

The training process involves optimizing for node classification and embedding the watermark. To obtain total complexity, we therefore need to consider two processes: forward passes with the GNN, and explaining the watermarked subgraphs.

**GNN Forward Pass Complexity.** The complexity of standard node classification in GNNs comes from two main processes: message passing across edges ($O(EF)$, where $E$ is number of edges and $F$ is number of node features), and weight multiplication for feature transformation ($O(NF^2)$, where $N$ is number of nodes). For $L$ layers, the time complexity of a forward pass is therefore:

$$O(L(EF + NF^2))$$

**Explanation Complexity.** Consider the Formula 3 for computing the explanation: $\mathbf{e} = explain(\mathbf{X}, \mathbf{P}) = (\tilde{K}^T\tilde{K} + \lambda I_F)^{-1}\tilde{K}^T\tilde{L}$. Remember that $\tilde{K}$ is an $N^2 \times F$ matrix, $I_F$ is a $F \times F$ matrix, and $\tilde{L}$ is a $N^2 \times 1$ vector. To compute the complexity of this computation, we need the complexity of each subsequent order of operations:

1. Multiplying $\tilde{K}^T\tilde{K}$ (an $O(F^2N^2)$ operation, resulting in an $F \times F$ matrix)
2. Obtaining and adding $\lambda I_F$ (an $O(F^2)$ operation, resulting in an $F \times F$ matrix)
3. Inverting the result (an $O(F^3)$ operation, resulting in an $F \times F$ matrix)
4. Multiplying by $\tilde{K}^T$ (an $O(F^2N^2)$ operation, resulting in an $F \times N^2$ matrix)
5. Multiplying the result by $\tilde{L}$ (an $O(F^2N^2)$ operation, resulting in an $N^2 \times 1$ vector)

The total complexity of a single explanation is therefore $O(F^2N^2) + O(F^2) + O(F^3) + O(F^2N^2) + O(F^2N^2) = O(F^2N^2 + F^3)$. For obtaining explanations of $T$ subgraphs during a given epoch of watermark embedding, the complexity is therefore:

$$O(T(F^2N^2 + F^3))$$

**Total Complexity.** The total time complexity over $i$ epochs is therefore:

$$O\left(i \times \left(L(EF + NF^2) + T(F^2N^2 + F^3)\right)\right)$$

### A.4 NORMALITY OF MATCHING INDICES DISTRIBUTION

Our results rely on the $z$-test to demonstrate the significance of the $MI$ metric. To confirm that this test is appropriate, we need to demonstrate that the $MI$ values follow a normal distribution. Table 2 shows the results of applying the Shapiro-Wilk Ghasemi & Zahediasl (2012) normality test to $MI$ distributions obtained under different GNN architectures and datasets. The results show $p$-values significantly above 0.1, indicating we cannot reject the null hypothesis of normality.

### A.5 ADDITIONAL RESULTS

**Fine-tuning and pruning under more GNN architectures.** The main paper mainly show results on GraphSAGE (Hamilton et al., 2018). Here, we also explore GCN Kipf & Welling (2017) and

| Dataset | SAGE | SGC | GCN |
|---------|------|-----|-----|
| Photo | 0.324 | 0.256 | 0.345 |
| CS | 0.249 | 0.240 | 0.205 |
| PubMed | 0.249 | 0.227 | 0.265 |

Table 2: Shapiro-Wilk Test p-values

| | | Number of Subgraphs ($T$) | | | | | | | | | | | | |
|---|---|---|---|---|---|---|---|---|---|---|---|---|---|---|
| | | | 2 | | | 3 | | | 4 | | | 5 | | |
| Dataset | GNN | Acc (Trn/Tst) | Wmk Align | MI $p$-val | Acc (Trn/Tst) | Wmk Align | MI $p$-val | Acc (Trn/Tst) | Wmk Align | MI $p$-val | Acc (Trn/Tst) | Wmk Align | MI $p$-val |
| **Photo** | GCN | 92.5/89.7 | 73.0 | 0.087 | 91.5/88.9 | 86.1 | <0.001 | 90.9/88.3 | 91.4 | <0.001 | 90.6/88.2 | 95.2 | <0.001 |
| | SGC | 92.0/89.4 | 73.8 | 0.111 | 91.0/88.7 | 82.5 | <0.001 | 90.1/88.0 | 91.8 | <0.001 | 89.7/87.4 | 99.4 | <0.001 |
| | SAGE | 95.4/88.9 | 77.4 | 0.002 | 94.4/87.5 | 90.9 | <0.001 | 94.1/88.2 | 97.7 | <0.001 | 93.9/87.2 | 99.4 | <0.001 |
| **PubMed** | GCN | 87.0/83.7 | 75.4 | 0.003 | 85.9/82.1 | 86.6 | <0.001 | 85.7/81.4 | 91.5 | <0.001 | 85.6/81.4 | 90.2 | <0.001 |
| | SGC | 86.7/83.1 | 79.7 | <0.001 | 85.8/81.6 | 83.8 | <0.001 | 85.3/81.4 | 88.9 | <0.001 | 84.6/80.0 | 92.9 | <0.001 |
| | SAGE | 91.9/82.8 | 76.8 | 0.009 | 91.3/81.8 | 81.0 | <0.001 | 91.1/81.2 | 85.2 | <0.001 | 90.1/79.6 | 91.5 | <0.001 |
| **CS** | GCN | 97.1/90.3 | 56.8 | 0.562 | 96.8/89.9 | 67.5 | <0.001 | 96.8/89.8 | 73.8 | <0.001 | 96.9/90.0 | 78.9 | <0.001 |
| | SGC | 97.2/90.3 | 57.1 | 0.003 | 96.8/89.9 | 67.7 | <0.001 | 96.7/90.1 | 74.5 | <0.001 | 96.6/89.8 | 77.8 | <0.001 |
| | SAGE | 99.9/90.2 | 61.5 | 0.233 | 99.9/89.4 | 73.3 | <0.001 | 99.9/88.9 | 78.2 | <0.001 | 99.9/88.3 | 84.0 | <0.001 |

Table 3: Watermarking results for varied $T$. Each value averages 5 trials with distinct random seeds.

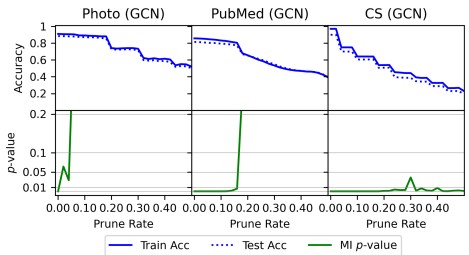

Figure 6: Pruning GCN models.

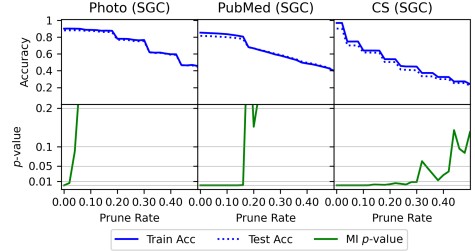

Figure 7: Pruning SGC Models.

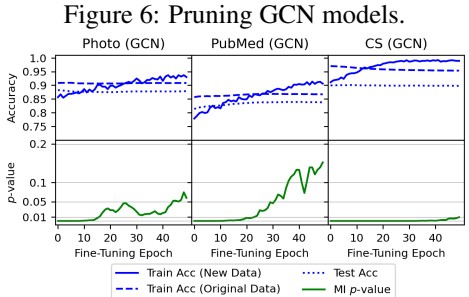

Figure 8: Fine-tuned GCN models.

Figure 9: Fine-tuned SGC models.

SGC (Wu et al., 2019). Figure 6-Figure 9 shows the impact of fine-tuning and pruning attacks results on our watermarking method under these two architectures. Watermarked GCN and SGC models fared well against fine-tuning attacks for the Photo and CS datasets, but less so for PubMed; meanwhile, these models were robust against pruning attacks for Pubmed and CS datasets, but not Photo. Since the owner can assess performance against these removal attacks prior to deploying their model, they can simply a matter of training each type as effectively as possible and choosing the best option. In our case, GraphSAGE fared best for our three datasets, but GCN and SGC were viable solutions in some cases.

**More Results on Effectiveness and Uniqueness.** Table 1 in the main paper shows the test accuracy, watermark alignment, and MI $p$-values of our experiments with the default value of $T = 4$. In Table 3, we additionally present the results for $T = 2$, $T = 3$, and $T = 5$. The results show MI $p$-values below 0.001 across all configurations when $T \geq 3$. They also show increasing watermark alignment

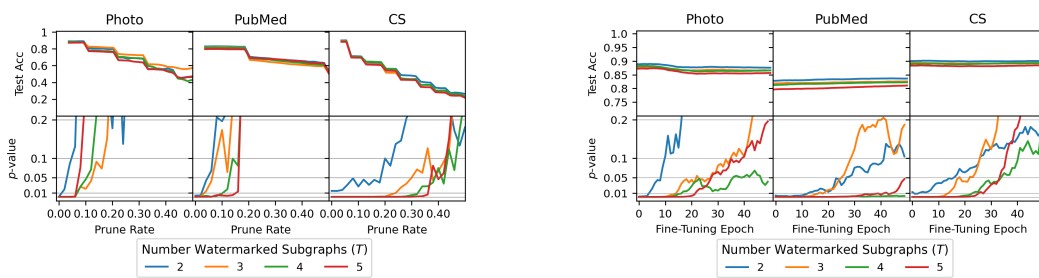

Figure 10: Pruning and fine-tuning attacks against varied number of watermarked subgraphs ($T$)

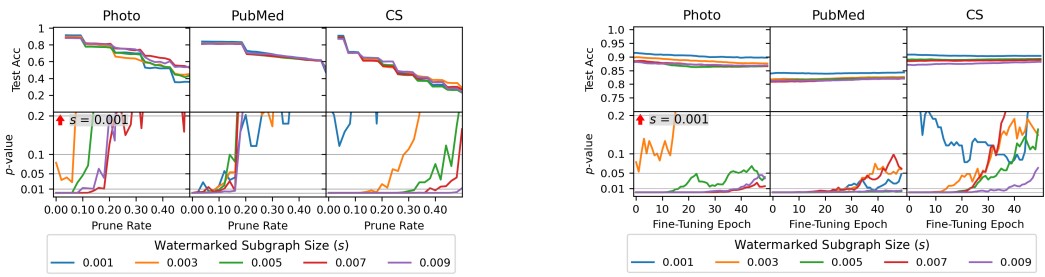

Figure 11: Pruning and fine-tuning attacks against varied sizes of watermarked subgraphs ($s$)

with increasing $T$, however, with a slight trade-off in classification accuracy: when increasing from $T = 2$ to $T = 5$, watermark alignment increases, but train and test classification accuracy decreases by an average of 1.44% and 2.13%, respectively; despite this, both train and test classification accuracy are generally high across all datasets and models.

**Fine-Tuning and Pruning under varied watermark sizes.** Figures 10 and 11 show the robustness of our methods to fine-tuning and pruning removal attacks when $T$ and $s$ are varied. We observe that, for $T \geq 4$ and $s \geq 0.005$ — our default values — pruning only affects MI $p$-value after classification accuracy has already been affected; at this point the pruning attack would be detected by model owners regardless. Similarly, across all datasets, for $T \geq 4$ and $s \geq 0.005$, our method demonstrates robustness against the fine-tuning attack for at least 25 epochs.

**Fine-Tuning under varied learning rates.** Our main fine-tuning results (see Figure 2) scale the learning rate to 0.1 times its original training value. Figure 12 additionally shows results for learning rates scaled to 1× and 10× the original training rates. The results for scaling the learning rate by 1× show that larger learning rates quickly remove the watermark. However, these figures also demonstrate that, by the time training accuracy on the fine-tuning dataset has reached an acceptable level of accuracy, the accuracy on the original training set drops significantly, which diminishes the usefulness of the fine-tuned model on the original task. For larger rates (10×), the watermark is removed almost immediately, but the learning trends and overall utility of the model are so unstable that the model is rendered useless. Given this new information, our default choice to fine-tune at 0.1× the original learning rate is the most reasonable scenario to consider.

### A.6 FUTURE DIRECTIONS.

**Extension to Other Graph Learning Tasks.**

While we have primarily provided results for the node-classification case, we believe much of our logic can be extended to other graph learning tasks, including edge classification and graph classification. Our method embeds the watermark into explanations of predictions on various graph features. Specifically, for node predictions, we obtain feature attribution vectors for the $n \times F$ node feature matrices of $T$ target subgraphs, with a loss function that penalizes deviations from the watermark. This process can be adapted to edge and graph classification tasks as long as we can derive $T$ separate $n \times F$ feature matrices, where $n$ represents the number of samples per group and $F$ corresponds to the number of features for the given data structure (e.g., node, edge, or graph). Below, we outline how this extension applies to different classification tasks:

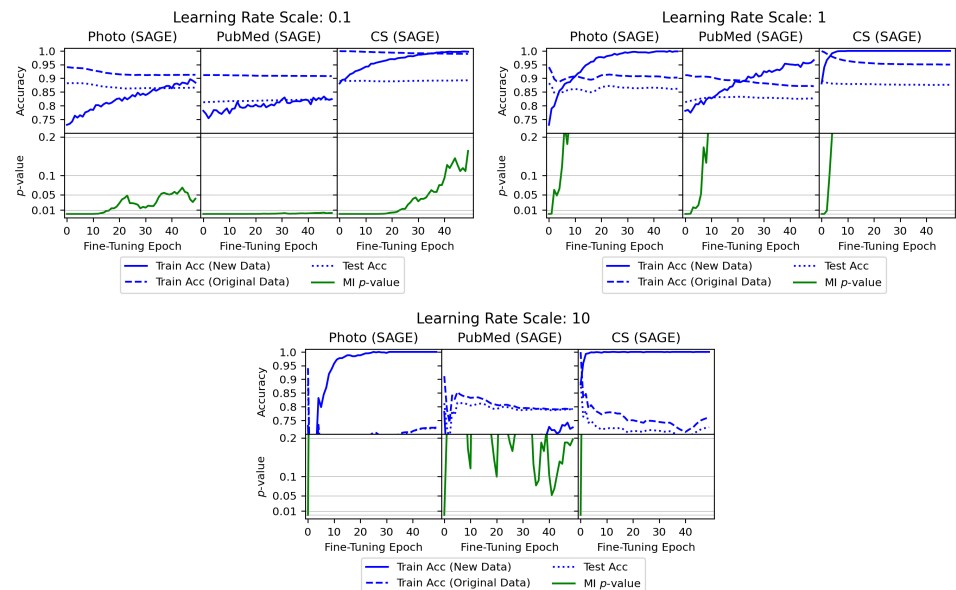

Figure 12: Fine-tuning results at increased learning rates (SAGE architecture).

1. **Node Classification:** The dataset is a single graph. Subgraphs are formed by randomly selecting $n = s \cdot |\mathcal{V}^{tr}|$ nodes from the training set (where $|\mathcal{V}^{tr}|$ is the number of training nodes and $s$ is a proportion of that size). (Note: in this case, $n$ is equal to the value $n_{sub}$ referenced previously in the paper.) For each subgraph:

   - The $n \times F$ node feature matrix represents the input features ($F$ is the number of node features).
   - The $n \times 1$ prediction vector contains one label per node.
   - These inputs are used in a ridge regression problem to produce a feature attribution vector for the subgraph.
   - With $T$ subgraphs, we generate $T$ explanations.

2. **Edge Classification:** Again, the dataset is a single graph. Subgraphs are formed by randomly selecting $n = s \cdot |\mathcal{E}^{tr}|$ edges. For each subgraph:

   - The $n \times F$ edge feature matrix represents the input features ($F$ is the number of edge features).
   - The $n \times 1$ prediction vector contains one label per edge.
   - These inputs are used in a ridge regression problem to produce a feature attribution vector for the subgraph.
   - As with node classification, we generate $T$ explanations for $T$ subgraphs.

3. **Graph Classification:** For graph-level predictions, the dataset $\mathcal{D}^{tr}$ is a collection of graphs. We extend the above pattern to $T$ collections of $n = s \cdot |\mathcal{D}^{tr}|$ subgraphs, where each subgraph is drawn from a different graph in the training set. Specifically:

   - Each subgraph in a collection is summarized by a feature vector of length $F$ (e.g., by averaging its node or edge features).
   - For a collection of $n$ subgraphs, we construct:
     - An $n \times F$ subgraph feature matrix, where each row corresponds to a subgraph in the collection.
     - An $n \times 1$ prediction vector, containing one prediction per subgraph.
   - These inputs are used in a ridge regression problem to produce a feature attribution vector for the collection.
   - With $T$ collections of $n$ subgraphs, we produce $T$ explanations.

By consistently framing each task as $T$ groups of $n \times F$ data points, our method provides a unified approach while adapting $F$ to the specific task requirements.

Table 4 shows sample results from applying the above framework to graph classification. These results are obtained using MUTAG, a 2-class dataset consisting of 188 chemical compounds that are labeled according to their mutagenic effects Debnath et al. (1991). We used the SAGE architecture with 3 layers. Each subgraph consists of 10 nodes, and each subgraph collection consists of 5 subgraphs. Each $1 \times F$ subgraph feature vector is obtained by averaging their node feature matrices over rows.

The results show that the MI $p$-value consistently remains below 0.05 for 4, 5, and 6 subgraph collections, demonstrating our method's effectiveness and beyond the node classification domain.

| | # Subgraph Collections | | |
|---|---|---|---|
| | **4** | **5** | **6** |
| **p-value** | 0.039 | 0.037 | <0.001 |
| **Acc (train/test)** | 0.915/0.900 | 0.954/0.929 | 0.915/0.893 |

Table 4: Watermarking results: graph classification

**Enhancing Robustness.**

An important future direction is to safeguard our method against model extraction attacks Shen et al. (2022), which threaten to steal a model's functionality without preserving the watermark. One form of model extraction attack is knowledge distillation attack Gou et al. (2020).

Knowledge distillation has two models: the original "teacher" model, and an untrained "student" model. During each epoch, the student model is trained on two objectives: (1) correctly classify the provided input, and (2) mimic the teacher model by mapping inputs to the teachers' predictions. The student therefore learns to map inputs to the teacher's "soft label" outputs (probability distributions) alongside the original hard labels; this guided learning process leverages the richer information in the teacher's soft label outputs, which capture nuanced relationships between classes that hard labels cannot provide. By focusing on these relationships, the student model can generalize more efficiently and achieve comparable performance to the teacher with a smaller model and fewer parameters, thus reducing complexity.

We find that in the absence of a strategically-designed defense, the knowledge distillation attack successfully removes our watermark ($p > 0.05$). This is unsurprising, since model distillation maps inputs to outputs but ignores mechanisms that lead to auxiliary tasks like watermarking.

To counter this, we outline a defense framework that would incorporate watermark robustness to knowledge distillation directly into the training process. Specifically, during training and watermark embedding, an additional loss term would penalize reductions in watermark performance. At periodic intervals (e.g., after every x epochs), the current model would be distilled into a new model, and the watermark performance on this distilled model would be evaluated. If the watermark performance (measured by the number of matching indices) on the distilled model is lower than the watermark performance on the main model, a penalty would be added to the loss term. This would ensure that the trained model retains robust watermarking capabilities even against knowledge distillation attacks.

