# OpenReview forum: "WATERMARKING GRAPH NEURAL NETWORKS VIA EXPLANATIONS FOR OWNERSHIP PROTECTION"
_ICLR.cc/2025/Conference — Submitted to ICLR 2025_

### Official Review · Reviewer_bM1q · 2024-11-03

**Soundness:** 2
**Presentation:** 3
**Contribution:** 2
**Rating:** 5
**Confidence:** 5

**Summary:**

The paper proposes a graph neural network watermarking method that embeds ownership information into the explanations of predictions. During the training process, the node classification loss maintains the model's classification accuracy, while the watermark embedding loss ensures that the feature explanations of the watermarked subgraphs align with the watermark pattern. After training, the model owner can claim its ownership by the similarity between the explanations and the watermark pattern. The author proves that it is an NP-hard problem for attackers to find the watermarked subgraphs. The method mitigates the impact of intentional misclassification compared to traditional backdoor-based watermarking techniques. Experiments on three datasets in node classification task illustrate the effectiveness of ownership verification and its robustness to pruning and fine-tuning.

**Strengths:**

The paper first introduces node feature explanations into GNN watermarking and demonstrates the potential of model explanations for applications in the GNN watermarking domain. By embedding the watermark within GNN prediction explanations rather than in the training data, the method mitigates risks associated with data pollution and misclassification.

**Weaknesses:**

1. Several prior methods have been proposed in the area of GNN watermarking, such as [1] and [2]. It is important to consider experiments that compare the watermark accuracy and downstream task performance of these methods.
2. The paper highlights the robustness of the method against pruning and fine-tuning. Given that adversaries may attempt model extraction attacks, as described in [3], it is advisable to address the robustness against such extraction attacks.

[1] Zhao, Xiangyu, Hanzhou Wu, and Xinpeng Zhang. "Watermarking graph neural networks by random graphs." 2021 9th International Symposium on Digital Forensics and Security (ISDFS). IEEE, 2021.

[2]Xu, Jing, et al. "Watermarking graph neural networks based on backdoor attacks." 2023 IEEE 8th European Symposium on Security and Privacy (EuroS&P). IEEE, 2023.

[3]Shen, Yun, et al. "Model stealing attacks against inductive graph neural networks." 2022 IEEE Symposium on Security and Privacy (SP). IEEE, 2022.

**Questions:**

1. How does the model perform without watermarking? It would be helpful to compare the classification accuracy on downstream tasks before and after embedding the watermark.
2. The explanation method GraphLIME is applied only to the node classification task in the paper. Can this watermarking method be extended to other downstream tasks in graph-based applications?

---

> ### Author Response · Authors · 2024-11-22
> **Response to Reviewer bM1q**
>
> Thank you for your response. We aim to address each of your comments below.
>
> # **Comparison to other GNN watermarking methods.**
> In our paper, we highlight the limitations of backdoor watermarking methods, particularly their introduction of intentional misclassification and ambiguity. Additionally, the metrics for evaluating watermarking success differ fundamentally: prior methods, such as [1] and [2], measure watermarking accuracy (prediction success on backdoor samples), while our approach relies on hypothesis testing to demonstrate statistically significant patterns in explanations. This makes direct comparisons impossible.
>
> Moreover, we emphasize that our method does not claim superiority in terms of classification accuracy. Instead, the key advantage of our method lies in addressing critical security concerns, such as the susceptibility of backdoor-based methods to data poisoning and the inherent ambiguity in watermark verification.
>
> # **Model extraction attack.**
>
> To address concerns about model extraction attacks, we implemented a knowledge distillation attack [3]. Knowledge distillation has two models: the original “teacher” model, and an untrained “student” model. During each epoch, the student model is trained on two objectives: (1) correctly classify the provided input, and (2) mimic the teacher model by mapping inputs to the teachers’ predictions. The student therefore learns to map inputs to the teacher’s “soft label” outputs (probability distributions) alongside the original hard labels; this guided learning process leverages the richer information in the teacher’s soft label outputs, which capture nuanced relationships between classes that hard labels cannot provide. By focusing on these relationships, the student model can generalize more efficiently and achieve comparable performance to the teacher with a smaller model and fewer parameters, thus reducing complexity.
>
> We find that in the absence of a strategically-designed defense, the knowledge distillation attack successfully removes our watermark ($p>0.05$). This is unsurprising, since model distillation maps inputs to outputs but ignores mechanisms that lead to auxiliary tasks like watermarking.
>
> To counter this, we outline a defense framework that would incorporate watermark robustness to knowledge distillation directly into the training process. Specifically, during training and watermark embedding, an additional loss term would penalize reductions in watermark performance. At periodic intervals (e.g., after every x epochs), the current model would be distilled into a new model, and the watermark performance on this distilled model would be evaluated. If the watermark performance (measured by the number of matching indices) on the distilled model is lower than the watermark performance on the main model, a penalty would be added to the loss term. This would ensure that the trained model retains robust watermarking capabilities even against knowledge distillation attacks.
>
> # **Performance without watermarking.**
>
> We have updated the paper (Table 1) to include accuracy rates in the absence of watermarking. For your convenience, we list them here. Note that accuracy rates are reasonable both with and without watermarking.
> **Accuracy (Train/Test)**|||||||
> |-|-|-|-|-|-|-|
> ||***GCN***||***SGC***||***SAGE***|
> |**Dataset**|No Watermark|Watermark|No Watermark|Watermark|No Watermark|Watermark|
> |***Photo***|91.3/89.4|90.9/88.3|91.4/89.9|90.1/88.0|94.2/90.8|94.1/88.2|
> |***PubMed***|88.6/85.8|85.7/81.4|88.8/85.9|85.3/81.4|90.5/86.0|91.1/81.2|
> |***CS***|98.5/90.3|96.8/89.8|98.4/90.3|96.7/90.1|100.0/88.4|99.9/88.9|
>
> # **Other graph tasks.**
>
> While our results focus on node classification, our method can be extended to other graph learning tasks, such as edge classification and graph classification. The key is to obtain $n \times F$ features matrices that we can derive explanations from. For node classification, each of our $T$ target subgraphs yields a $n \times F$ matrix of node features. For edge classification, we can generate similar matrices for edge features. For graph classification, we can accomplish something similar by having each row in a $n \times F$ matrix represent the features of an individual graph; rather than have one such matrix for each of $T$ target subgraphs, we instead have one matrix per *collection* of subgraphs.
>
>
> Please see further details of this extended logic in Appendix Section A.5 in our revisions.
>
> ---
>
> ## References
>
> [1] Zhao, Xiangyu, Hanzhou Wu, and Xinpeng Zhang. "Watermarking graph neural networks by random graphs." 2021 9th International Symposium on Digital Forensics and Security (ISDFS). IEEE, 2021.
>
> [2] Xu, Jing, et al. "Watermarking graph neural networks based on backdoor attacks." 2023 IEEE 8th European Symposium on Security and Privacy (EuroS&P). IEEE, 2023.
>
> [3] Gou, Jianping et al. “Knowledge Distillation: A Survey.” International Journal of Computer Vision 129 (2020): 1789 - 1819.

---

> ### Author Response · Authors · 2024-11-25
> **Update: Extension to other Graph Learning Tasks**
>
> # Update: Extension to other Graph Learning Tasks
>
> Thank you again for your previous comments on our work. Your question about applications to other graph learning tasks was mirrored by Reviewer ub6g, and to address this, we now have additional results to share.
>
> In our previous comments, we outlined a framework for extending our methodology to other graph learning tasks (more details in Appendix Section A.5). To illustrate the broader applicability of our approach, we have now included results from watermarking explanations of predictions on the widely-used MUTAG [1] dataset within the graph classification framework. These results consistently yield p-values below 0.05 under varied configurations, demonstrating that our method can be effectively applied to other graph learning tasks.
>
> ||# Collections |of Subgraphs||
> |-|-|-|-|
> ||4|5|6||
> |**p-value**|0.039|0.037|<0.001||
> |**Acc (train/test)**|0.915/0.900|0.954/0.929|0.915/0.893|
>
> While the edge classification framework would align closely with our node classification approach, watermarking for graph classification requires more modifications to our framework; therefore, the above results provide strong evidence of the flexibility of our method.
>
> ---
>
> [1] Debnath, Asim Kumar et al. “Structure-activity relationship of mutagenic aromatic and heteroaromatic nitro compounds. Correlation with molecular orbital energies and hydrophobicity.” Journal of medicinal chemistry 34 2 (1991): 786-97.

---

### Official Review · Reviewer_7pbz · 2024-11-04

**Soundness:** 3
**Presentation:** 2
**Contribution:** 2
**Rating:** 5
**Confidence:** 3

**Summary:**

This paper aims to inject watermarks to graph neural networks, in order to serve as a verification strategy to protect the ownership of the graph neural network. In order to achieve the goal, the authors investigate a new methodology of graph model explanation.

**Strengths:**

1. The paper studies a novel problem for GNN ownership verification.
2. The proposed method is interesting and the empirical results demonstrate the effectiveness.
3. The authors conduct serious robustness evaluation of the proposed methodology.

**Weaknesses:**

**Concerns Regarding the Presentation**\
From Section 3, the paper uses too many math notations and equations, but didn't provide sufficient explanation and clarification. This make the paper not very easy to follow. For example, the GNN explanation part in Section 3.2, it is hard to see how someone can use it to explain the GNN prediction. Moreover, the switch from Lasso to Ridge Regression will make the give weights to all features. I am not sure how to find important features based on the Ridge Regression output.

**Concerns Regarding the Threat Model**\
The threat model discussed in this paper appears to lack sufficient justification. Specifically:
1. It is unclear why or under what circumstances “the adversary does not know the location of the watermarked subgraphs, but knows the shape and number of the subgraphs.” In practice, if the adversary is not the model trainer, they may have limited knowledge about the watermark.
2. The authors also mention that the model ownership verifier cannot access the protected model. It’s not immediately clear why the verifier would be unable to access the model parameters. Try to imagine, if an artist who owns a painting work sues another party for copyright infringement, it would be unusual for the artist to prevent the judges from examining their original artworks.

**Questions:**

Plz see the weakness.

---

> ### Author Response · Authors · 2024-11-22
> **Response to Reviewer 7pbz**
>
> Thank you for your thoughtful review and comments. We address your concerns below.
>
> # **GNN explanation intuition.**
>
> To streamline our description, we have moved some of the detailed math in Section 3.2 to Appendix A.2, keeping only the final regression problem in the main paper. Here, we provide high-level insight into how the explanation process works:
>
> - Our explanations are the output of a closed-form ridge regression problem applied to a transformation of the node features and node predictions (using Gaussian kernel matrices inspired by GraphLIME, though we do not use GraphLIME itself).
> - While ridge regression assigns non-zero weights to all features, the relative magnitudes of the regression coefficients indicate the most influential features. However, our goal is not to analyze the explanations themselves for feature importance. Instead, we use the explanations as an embedding space for our watermark.
> - During our embedding process, we intentionally manipulate the feature attribution vectors to align with a predefined watermark, prioritizing the ability to match a specific pattern over interpretability. Ridge regression’s non-zero weights  provide the flexibility needed for precise alignment with the desired watermark. By contrast, Lasso regression’s sparsity-inducing nature would limit the ability to manipulate attribution vectors for embedding, making it less effective for our purpose.
>
> To clarify, our method does not rely on the explanations to interpret GNN predictions. Instead, we use them as a structured space where ownership information can be securely embedded and later verified.
>
> # **Adversary knowledge.**
>
> Our primary assumption is that the adversary does not have knowledge about the watermarked subgraphs. While we consider scenarios where they might know additional details, such as the number and size of the subgraphs, this is not our main focus. Rather, we include this analysis to demonstrate that even with such additional knowledge, our method remains robust. These scenarios are possible but not central to our default assumptions; they are presented to emphasize the resilience of our approach under varying conditions.
>
> # **Verifier knowledge.**
>
> Our point in stating that the verifier only has black-box access is to emphasize that no additional information or model access is necessary for verification. Even in a scenario where the verifier is restricted to this limited level of access, they can still successfully perform the verification task. This highlights the robustness of our approach, which does not rely on access to model parameters or other privileged information.

---

### Official Review · Reviewer_k1GL · 2024-11-08

**Soundness:** 3
**Presentation:** 2
**Contribution:** 3
**Rating:** 5
**Confidence:** 3

**Summary:**

This paper proposes a graph explanation-based watermarking method. The watermarks are set as specific explanations and are trained into the protected model by backdoor techniques. During verification, explanations of watermarked data will be compared with the ground-truth explanations to determine the ownership. In empirical evaluations, authors demonstrate the performance of the proposed watermarking method from different views, e.g., effectiveness, uniqueness, robustness, and undetectability.

**Strengths:**

It is novel to utilize explanation as 'poisoned labels' of watermarked data, which will not negatively influence the performance of GNNs.

**Weaknesses:**

The major concern is how verifiers could get the explanations via black-box access to the protected model during the black-box verification. If I read carefully enough, the explanation of GraphLIME can only be obtained under white-box settings with node embeddings or other model parameters. How is the query efficient if they can be obtained via black-box access? I welcome discussion from authors and will consider raising the score if the answer is appropriate.

**Questions:**

Please refer to weaknesses.

---

> ### Author Response · Authors · 2024-11-22
> **Response to Reviewer k1GL**
>
> Thank you for your question. We address your question below.
>
> # Black-Box Explanations and Verification: Clarification
>
> It is true that GraphLIME itself is an iterative, gradient-based process that requires *white-box* model access. To clarify, we do not directly apply GraphLIME: rather, the Gaussian kernel matrices used by GraphLIME inspire our closed-form regression problem. (We've revised our Introduction and Section 3.2 to emphasize this point.) This process takes node features and node predictions as inputs and outputs a feature attribution vector. Notably, it operates in a *black-box setting*, meaning it does not require access to the protected model's internal parameters. This query is highly-efficient, since the closed-form regression only requires a single query.
>
> We welcome further questions on this topic.

---

> > ### Comment · Reviewer_k1GL · 2024-11-26
> > **Response to submission 2714**
> >
> > I sincerely appreciate the authors' responses and read other reviewers' comments. However, this paper requires many revisions, which may not be allowed by ICLR. Therefore, I will not vote for acceptance.

---

> > > ### Author Response · Authors · 2024-11-27
> > > **Revisions Summary**
> > >
> > > Thank you for your feedback. We'd like to summarize the revisions to our paper. (These changes are highlighted in the current revision in blue. Note: the majority of these changes are in the appendix.)
> > >
> > > **Clarifications**
> > > - Added references to [1] in the Introduction, Related Works, and Section 4.1 to better contextualize our contributions.
> > > - Moved detailed mathematical derivations from Section 3.2 to Appendix A.2 to streamline our main text.
> > > - Clarified GraphLIME’s relation to our explanation method.
> > >
> > > **Experiments**
> > > - Added a non-watermark accuracy column to Table 1.
> > > - Tested the distribution of match indices for normality to ensure the validity of our statistical approach (Appendix Table 2).
> > > - Corrected an error in our fine-tuning results in Figure 2 and included additional fine-tuning results (Appendix Figure 12).
> > > - Obtained preliminary results extending our method to graph classification (Appendix Table 4).
> > >
> > > **Future Work**
> > > - Outlined potential extensions of our method to other graph learning tasks and defenses against model distillation attacks, highlighting its versatility and adaptability (Appendix A.6)
> > >
> > > While these changes address reviewer feedback and improve the paper’s clarity and robustness, they do not alter the core methodology or main contributions of our work. Instead, they provide additional evidence to support our claims, improve presentation, and demonstrate the method’s broader scope and applicability. These refinements enhance the paper’s overall quality without shifting its scope or focus.
> > >
> > > ---
> > >
> > >
> > > [1] Shao, Shuo, Yiming Li, Hongwei Yao, Yiling He, Zhan Qin, and Kui Ren. “Explanation as a Watermark: Towards Harmless and Multi-bit Model Ownership Verification via Watermarking Feature Attribution.” Proceedings of the Network and Distributed System Security Symposium (NDSS), 2025.

---

### Official Review · Reviewer_ub6g · 2024-11-08

**Soundness:** 3
**Presentation:** 3
**Contribution:** 3
**Rating:** 5
**Confidence:** 2

**Summary:**

In this paper, the authors propose to watermarking GNNs via the explanations of GNNs. Their theory indicates that locating the watermark is an NP-hard problem and the experiments demonstrate that the proposed attack is robust to current defenses.

**Strengths:**

1 This motivation of the paper is clear.

2 The experiments are quite solid.

3 The soundness of the proposed method is good.

**Weaknesses:**

1 Although the authors claim that the proposed method can avoid data pollution and eliminates intentional misclassification. However, compared to the traditional methods, it requires a more tight threat model: the training process of target models. It is not always accessible to all scenarios.

2 In Figure 2, the results indicate that fine-tuning fails to remove the effectiveness of the embedded watermark. However, as far as I know, the effectiveness of the fine-tuning attack is closely related to the detailed setting of the learning rate, A larger learning rate might attack the proposed method.

3 In this paper, the authors only narrows the attack in the node classfication. It is only a sub-task in the graph community. I am not sure whether the attack is general enough to be applied to another task, such as edge classification.

**Questions:**

1 As far as I know, GNNs is only one type of models for node classification, how about more powerful models such as graph transformers?

2 In Table 1, the authors only show the accuracy after inserting watermark. How about accuracy with performing watermarking? Those data is needed to demonstrate the stealthiness of the attack.

---

> ### Author Response · Authors · 2024-11-22
> **Response to Reviewer ub6g**
>
> Thank you for your comments! We address each of your concerns below.
>
> # **Tight threat model.**
>
> We’d like to clarify the capabilities of each party in our threat model.
>
> - ***Adversaries:*** We assume adversaries have black-box access to the model and no knowledge of the watermark. Even in cases where adversaries gain white-box access (e.g., for fine-tuning or pruning), locating the watermark remains computationally intractable without precise knowledge of its location.
>
> - ***Verifier:*** The verifier only requires black-box access to the model for verification, making the approach broadly applicable and robust.
>
> - ***Model Owner:*** The model owner has white-box access to the training process and the data. This assumption is realistic and not unique to our method; many backdoor-based watermarking approaches also rely on access to training data for embedding or verification purposes [1-3].
>
> Our method minimizes reliance on data pollution and avoids ambiguity, addressing the limitations of backdoor-based watermarking methods while leveraging access commonly assumed in this domain. We believe this combination of practicality and robustness makes our approach suitable for a wide range of scenarios, even if universal applicability is not guaranteed.
>
> # **Fine-tuning with a larger learning rate.**
>
> Our default choice for fine-tuning learning rate was 0.1x the learning rate on the original task. To address your concern, we have conducted additional fine-tuning experiments with scaled learning rates (1x and 10x) – see Figure 12 in the appendix of our revised paper.
>
> We find that larger learning rates accelerate the rise in the MI p-value, indicating faster degradation of the watermark. For instance, at 1x, the p-value increases sharply within the first 10 epochs. However, this comes at a significant cost: by the time fine-tuning accuracy (on the new dataset) increases to a reasonable level, accuracy on the *original* training set drops substantially, making the attack impractical as the model’s utility on the original task is severely compromised.
>
> Additionally, the fine-tuning results in the original submission were inadvertently obtained by fine-tuning on the original training dataset. In the updated experiments, we fine-tuned on a separate validation set to better reflect realistic attack scenarios. This adjustment introduced some differences—most notably, a decline in original training accuracy—but, as previously noted, this strengthens our position by showing that fine-tuning attacks not only affect the watermark but also significantly degrade the model’s utility, reducing the attack’s stealth. Despite these changes, the p-value trends remain consistent: for Photo and PubMed datasets, the p-value stays roughly below 0.1 throughout fine-tuning, and for CS, it only rises above 0.1 after prolonged fine-tuning. These findings reaffirm the robustness of our method while highlighting the trade-offs faced by an attacker.
>
>
> # **Other graph tasks.**
>
> While our results focus on node classification, our method can be extended to other graph learning tasks, such as edge classification and graph classification. The key is to obtain $n \times F$ features matrices that we can derive explanations from. For node classification, each of our $T$ target subgraphs yields a $n \times F$ matrix of node features. For edge classification, we can generate similar matrices for edge features. For graph classification, we can accomplish something similar by having each row in a $n \times F$ matrix represent the features of an individual graph; rather than have one such matrix for each of $T$ target subgraphs, we instead have one matrix per *collection* of subgraphs.
>
> Please see further details of this extended logic in Appendix Section A.5 in our revisions.
>
> # **Other models.**
>
> Graph Neural Networks are the more popular choice for graph learning tasks and a natural starting point for node classification. In contrast, Graph Transformers introduce significant computational complexity due to their reliance on global attention mechanisms [4], adding unnecessary overhead to the multi-part optimization task. While extending our method to Graph Transformers is an interesting direction for future work, it falls outside the current scope of our focus.
>
> # **Accuracy without watermarking.**
>
> We have updated the paper (Table 1) to include accuracy rates in the absence of watermarking. For your convenience, we list them here. Note that accuracy rates are reasonable both with and without watermarking.
>
> |**Accuracy (Train/Test)**|||||||
> |-|-|-|-|-|-|-|
> ||***GCN***||***SGC***||***SAGE***|
> |**Dataset**|No Watermark|Watermark|No Watermark|Watermark|No Watermark|Watermark|
> |***Photo***|91.3/89.4|90.9/88.3|91.4/89.9|90.1/88.0|94.2/90.8|94.1/88.2|
> |***PubMed***|88.6/85.8|85.7/81.4|88.8/85.9|85.3/81.4|90.5/86.0|91.1/81.2|
> |***CS***|98.5/90.3|96.8/89.8|98.4/90.3|96.7/90.1|100.0/88.4|99.9/88.9|

---

> ### Author Response · Authors · 2024-11-22
> **References**
>
> [1] Zhao et al. “Watermarking Graph Neural Networks by Random Graphs.” ISDFS, 2020
>
> [2] Xu et al. “Watermarking Graph Neural Networks based on Backdoor Attacks.” EuroS&P 2023
>
> [3] Bansal, Arpit, et al. “Certified Neural Network Watermarks with Randomized Smoothing.” ICML 2022
>
> [4] Müller, Luis, Mikhail Galkin, Christopher Morris, and Ladislav Rampášek. “Attending to Graph Transformers.” TMLR, 2024.

---

### Public Comment · ~Yiming_Li1 · 2024-11-13
**Alert for Potential Plagiarism: Highly similar to our work without proper reference and discussions**

**Summary**: This paper proposed to watermark the explanations of Graph Neural Networks (GNN) to protect the copyright of GNNs. Specifically, this paper utilized GraphLIME to generate the explanations and embed the watermark into them via a dual-objective loss function. This paper further provided a theoretical analysis of the difficulty of locating the watermark.


**Soundness**: fair

**Presentation**: fair

**Contribution**: Poor



**Strengths**

1. The studied problem is of great significance and sufficient interest to ICLR audiences.
2. The main idea is easy to follow to a large extent.
3. The authors try to provide theoretical analyses of their method, which should be encouraged.



**Weaknesses**

1. This paper is very similar to [1] (initially submitted to S&P 2024 on December 7, 2023 and released on arXiv in May 8, 2024, link: https://arxiv.org/abs/2405.04825) in various aspects. It appears to be a straightforward implementation of [1] on Graph Neural Networks. However, it seems that the authors intentionally omit the reference to [1] and exaggerate their contributions by not citing [1].  The authors should provide a detailed discussion of the differences and originality of this paper compared with [1]. Otherwise, this paper might be regarded as plagiarism. The resemblances are outlined below.
   1. These two papers share **the same motivation** that existing backdoor-based watermarking methods are harmful and ambiguous.
   2. These two papers share **the same insight** to embed the watermark into the explanation of the neural network.
   3. These two papers share **nearly the same method** to embed the watermark. [1] proposed to utilize a LIME-based method, while this paper leverages GraphLIME (which is an extension of LIME in GNN).
   4. Both papers utilize **ridge regression** to calculate the explanations.
   5. Both papers use **hypothesis-based methods** for ownership verification.  [1] utilized the chi2-test while this paper uses the z-test.
2. This paper uses Z-test for ownership verification and the follow-up analyses, based on matching indices. However, Z-test can be used only when the variable follows the Gaussian distribution (https://en.wikipedia.org/wiki/Z-test). However, to the best of our knowledge, the matching indice (MI) follows a multinomial distribution instead of the Gaussian distribution. In this case, it is not appropriate to use the Z test. Even if the author tries to claim that the distribution of MI approaches the Gaussian distribution under the central limit theorem, the author needs to verify its correctness through the Normality Test.
3. Missing some important baselines [2, 3]. This paper also lacks an empirical comparison with these baselines.
4. The experiments on the robustness are inadequate. There is no discussion about the resistance to potential adaptive attacks. For example, the adversaries can design a strong adaptive attack by including the proposed method as a regularization and conducting overwirte attacks.



References:

1. Explanation as a Watermark: Towards Harmless and Multi-bit Model Ownership Verification via Watermarking Feature Attribution. NDSS, 2025.
2. Revisiting Black-box Ownership Verification for Graph Neural Networks. S&P, 2024.
3. Watermarking Graph Neural Networks by Random Graphs. ISDFS, 2021.


**Questions**
1. Clarify the differences and novelty of this paper compared with [1].
2. Provide empirical comparisons with existing baseline methods.

---
**Alert for Potential Plagiarism**

We have good reasons to believe that the core methodology of this paper was ported over from our NDSS'25 paper (Explanation as a Watermark: Towards Harmless and Multi-bit Model Ownership Verification via Watermarking Feature Attribution). The authors exaggerate their contribution in this submission by deliberately not citing our NDSS paper. In general, our paper provides a general watermarking method that can be used for both image classification and text generation. This paper is just an implementation of our paper in graph classification, although we did not do GNN experiments in our paper.

Although this paper was accepted in September 2024, and I know ICLR has a 3-month concurrent work policy. However, our work was uploaded to arxiv in early May this year (https://arxiv.org/abs/2405.04825), not to mention that it had already been reviewed by two conferences (SP, CCS). This paper is not only similar to ours in the general direction but also completely plagiarizes our technology. We believe that the original intention of the concurrent work policy is to protect authors from missing comparative work due to objective reasons, rather than to protect malicious plagiarism due to subjective intention.

I urge the authors to explain this issue and provide a proper and comprehensive illustration and comparison of our work.

---

> ### Author Response · Authors · 2024-11-17
> **Comparison of Our Method to EaaW**
>
> # Comparison to EaaW
> Thank you for the review and opportunity to address your concerns regarding our work’s relation to “Explanation as a Watermark” (EaaW) [1]. We address each of your key points below, outlining our contributions, differences, and similarities.
>
> ### **Contributions**
>
> - ***Watermarking in the graph domain.*** Our work fills a gap in the literature by extending explanation-based watermarking to the graph domain, requiring non-trivial adaptations for explanation generation and watermark alignment.
>
> - ***Verification without reliance on ground truth.***  Our method avoids requiring a third party to know the ground-truth watermark, enhancing security by eliminating the need to share private information.
>
> - ***Theoretical hardness of watermark identification.*** We prove that locating our watermarks is NP-hard, providing a formal guarantee of robustness.
>
> ### **Differences**
>
> - ***Domain-specific focus.***  Our work focuses on the graph domain, addressing structural relationships, multi-hop dependencies, and graph connectivity—challenges absent in the standard DNNs used in EaaW.
>
> - ***Beyond LIME-Based Sampling.*** Unlike GraphLIME [2], LIME [3], and EaaW, we do not rely on local sampling around individual inputs. Instead, our target subgraphs consist of multiple disconnected nodes sampled from across the node classification network, reducing overlap with adversary-selected subgraphs. Additionally, we watermark only a selected subset of node features rather than the full feature signal, enhancing robustness and minimizing watermark size. To clarify, GraphLIME’s primary influence on our work lies in its use of Gaussian kernel matrices for regression, as described in Section 3.2.
>
>  - ***Regression input.***  EaaW’s regression inputs are binary masks isolating specific feature subsets from a single trigger sample, enabling targeted attribution analysis. In contrast, our regression takes a matrix of node features aggregated across disconnected nodes, capturing broader feature relationships in the graph domain.
>
> - ***Disjoint training sets.*** EaaW optimizes for classification performance on a training set that includes the trigger sample. In contrast, our method uses disjoint subsets for classification and watermarking optimization; we found this separation was necessary for loss convergence.
>
> - ***Verification method.*** Our method does not compare adversarial samples to a ground-truth trigger, instead requiring target subgraphs with maximally similar explanations.
>
>
> ### **Similarities**
>
> - ***Motive and high-level insight.***
> Both works aim to address the limitations of backdoor-based watermarking by watermarking explanations. This does not undermine our originality, as the challenges of backdoor mechanisms are well-documented [4,5,6], providing a natural and necessary starting point to consider other embedding spaces, like explanations.
>
> - ***High-level embedding approach.*** Both EaaW and our work use ridge regression and a hinge-like loss function to embed watermarks. We acknowledge this similarity and will cite EaaW in section 4.1 of our paper. However, our methodology diverges in domain, sampling, and verification.
>
> ---
>
> We acknowledge that EaaW should have been cited. This omission was not intentional. Given the similar embedding approach, deliberately omitting citation would have been counterproductive; it was a mistake made during a time-constrained submission process, and we regret this oversight. While our embedding approaches overlap, our contributions lie in domain-specific adaptations, novel applications, and verification strategies. These distinctions will be clarified further in our revisions.
>
> ---
>
> ### References
>
> [1] Shao, Shuo, Yiming Li, Hongwei Yao, Yiling He, Zhan Qin, and Kui Ren. “Explanation as a Watermark: Towards Harmless and Multi-bit Model Ownership Verification via Watermarking Feature Attribution.” Proceedings of the Network and Distributed System Security Symposium (NDSS), 2025.
>
> [2] Huang, Q. et al. “GraphLIME: Local Interpretable Model Explanations for Graph Neural Networks.” IEEE Transactions on Knowledge and Data Engineering 35 (2020): 6968-6972.
>
> [3] Ribeiro, Marco Tulio et al. ““Why Should I Trust You?”: Explaining the Predictions of Any Classifier.” Proceedings of the 22nd ACM SIGKDD International Conference on Knowledge Discovery and Data Mining (2016): n. pag.
>
> [4] Gu, Tianyu et al. “BadNets: Evaluating Backdooring Attacks on Deep Neural Networks.” IEEE Access 7 (2019): 47230-47244.
>
> [5] Liu, Jian, Rui Zhang, et al. “False Claims against Model Ownership Resolution.” Proceedings of the USENIX Security Symposium, 13 Apr. 2023.
>
> [6] Yan, Yifan, et al. "Rethinking {White-Box} Watermarks on Deep Learning Models under Neural Structural Obfuscation." *Proceedings of the 32nd USENIX Security Symposium (USENIX Security 23)*, 2023, pp. 2347–2364.

---

> ### Author Response · Authors · 2024-11-17
> **Suggested Improvements**
>
> # Suggested Improvements
> To address concerns about normality, we applied the Shapiro-Wilk test [1] to our Matching Indices (MI) distributions. Below are the average p-values from 5 trials for three GNN architectures. These results fail to reject the null hypothesis of normality.
>
> |Shapiro-Wilk Test p-values||||
> |-|-|-|-|
> |Dataset|SAGE|SGC|GCN|
> |Photo|0.324|0.256|0.345|
> |CS|0.249|0.240|0.205|
> |PubMed|0.249|0.227|0.265|
>
> ### **Baselines and Robustness Experiments.**
> We aim to include these results as part of our broader revisions alongside other responses.
>
> ---
>
> ### References
>
> [1] Ghasemi, Asghar and Saleh Zahediasl. “Normality Tests for Statistical Analysis: A Guide for Non-Statisticians.” International Journal of Endocrinology and Metabolism 10 (2012): 486 - 489.

---

> ### Public Comment · ~Yiming_Li1 · 2024-11-21
>
> Dear Author(s):
>
> Thank you for your response and the revisions made to the paper. I appreciate that the revised version now includes a comparison with our previous NDSS paper, and I am pleased to see that the contributions based on our work have been fairly presented. Given the methodological similarities between the two works, I believe such a discussion is indeed essential.
>
> I also acknowledge the effort the authors have made in designing and implementing a watermarking solution specifically tailored for GNNs. As such, I respectfully request that the reviewers and area chair reconsider the contributions of this paper in light of the revised version.
>
> PS: While I agree that the reviewers partially addressed my concerns about the methodology in terms of hypothesis testing, I still have some doubts about the comprehensiveness of the paper in terms of robustness. I suggest the authors conduct more adaptive attacks for discussion.
>
> Best Regards,

---

> > ### Author Response · Authors · 2024-11-25
> >
> > Thank you for the follow-up and for acknowledging the revisions to our work; we are glad that you feel the revisions fairly represent your contributions. We appreciate the opportunity to clarify the independent value of our work: our application to graph neural networks, unique verification strategy, and theoretical guarantee stand out as unique contributions of our method.
> >
> > We believe that our current results sufficiently demonstrate the robustness of our method against various threats. Specifically, we show robustness against fine-tuning and pruning attacks, and we also prove that locating our watermark is NP-hard. That said, we recognize that exploring robustness against adaptive attacks could be valuable future work.
> >
> > Thank you again for your thoughtful comments, as they helped us improve our paper.

---

### Comment · Area_Chair_HF3v · 2024-11-24

Dear reviewers,

Thanks for serving as a reviewer. As the discussion period comes to a close and the authors have submitted their rebuttals, I kindly ask you to take a moment to review them and provide any final comments.

If you have already updated your comments, please disregard this message.

Thank you once again for your dedication to the OpenReview process.

Best,

Area Chair

---

### Meta-Review · Area_Chair_HF3v · 2024-12-19

**Metareview:**

The paper proposes a watermarking method for GNN based on the explanations on graphs. During the training, the work adds an additional watermarking embedding loss to ensure the explanations' features of the watermarked subgraphs aligning with the water pattern. Then the model owners can use the similarity between the model's explanations and the watermark pattern to claim their ownership. Experiments and theoretical analysis demonstrate their effectiveness of ownership verification and robustness to pruning and fine-tuning.

Strength:
1. The proposed method based on GNN's explanation is interesting and effective under their evaluation.

Weaknesses:
1. The experiment settings are limited and may be wired considering its threat model.

2. The work cannot work under a model extraction attack.

3. The writing for its method needs to be more clear.

As all reviewers are worried about the method's scope and effectiveness, I think this author should add more before this paper is accepted. I would like to propose some new comments for the authors of this paper:

1. I think the watermarking scenarios for GNN can only be valid when the GNN is complex or the training dataset is really large. Therefore, I suggest you conduct your experiments at least in the OGBN dataset. I think fine-tuning a small GNN trained on Pubmed, Photo, CS or MUTAG is not realistic, as they can be retrained easily with only a few minutes.

2. More complex GNNs should be considered, more commonly used GNNs like GAT, GCNII, APPNP, or even ResGCN should be considered to enhance the paper's impacts.

3. Add more experiments on other tasks like Graph classification.

**Additional Comments On Reviewer Discussion:**

All reviewers are worried about the method's scope and effectiveness. As for the plagiarism claim raised by the public comments, both reviewers and I do not agree. However, most reviewers and I agree that the authors should do more before this paper is accepted.

---

### Decision · Program_Chairs · 2025-01-22

Reject